# SAD-Flower: Flow Matching for Safe, Admissible, and Dynamically Consistent Planning

## Abstract

Flow matching (FM) has shown promising results in data-driven planning. However, it inherently lacks formal guarantees for ensuring state and action constraints, whose satisfaction is a fundamental and crucial requirement for the safety and admissibility of planned trajectories on various systems. Moreover, existing FM planners do not ensure the dynamical consistency, which potentially renders trajectories inexecutable. We address these shortcomings by proposing SAD-Flower, a novel framework for generating **S**afe, **A**dmissible, and **D**ynamically consistent trajectories. Our approach relies on an augmentation of the flow with a virtual control input. Thereby, principled guidance can be derived using techniques from nonlinear control theory, providing formal guarantees for state constraints, action constraints, and dynamic consistency. Crucially, SAD-Flower operates without retraining, enabling test-time satisfaction of unseen constraints. Through extensive experiments across several tasks, we demonstrate that SAD-Flower outperforms various generative-model-based baselines in ensuring constraint satisfaction.

## 1 Introduction

Generative models have recently emerged as powerful tools for trajectory planning, with diffusion models (Ho et al., 2020) and flow matching (FM) (Lipman et al., 2023) enabling the generation of complex, long-horizon behaviors by directly learning from data. Unlike traditional data-driven planners that combine learned dynamics with optimization routines (Posa et al., 2014; Kalakrishnan et al., 2011), generative approaches avoid model exploitation—where optimizers produce trajectories that perform well under the model but fail in reality due to approximation errors (Talvitie, 2014; Ke et al., 2019). By training models to generate full trajectories that implicitly encode system dynamics, generative planners naturally capture multimodal (Huang et al., 2025), high-dimensional behaviors while mitigating compounding errors and supporting task compositionality. These advantages make generative approaches increasingly attractive for real-world planning and control tasks.

Despite these advantages, a critical limitation of generative model planners lies in their inability to guarantee constraint satisfaction – specifically, state and action constraints. State constraints ensure safety (Wabersich et al., 2023) (e.g., avoiding collisions), while action constraints guarantee admissibility (Shen et al., 2018) (e.g., respecting torque or power limits), which makes these constraints essential in domains such as robotics (Craig, 2009). However, constraint satisfaction at individual time steps is insufficient: since trajectories are sequences of states (and actions), subsequent states in trajectories cannot be chosen independently (Kelly, 2017). For a planned trajectory to be physically realizable and its guarantees to transfer from plan to execution, the trajectory must be dynamically consistent, i.e., states must evolve according to the dynamics of the system for which the trajectory is planned. However, existing generative planners offer no inherent mechanism to enforce such properties, and the non-static nature of real-world environments often leads constraints to be underrepresented, or entirely absent, in the training data. These factors make constraint satisfaction particularly challenging at test time and motivate the need for formally grounded methods that can reliably ensure safety, admissibility, and consistency in generated trajectories.

A number of recent works have explored constrained planning for generative models by incorporating constraints into the training phase (Ho & Salimans, 2022; Ajay et al., 2023; Zheng et al., 2023), injecting guidance signals during sampling (Dhariwal & Nichol, 2021; Yuan et al., 2023; Kondo et al., 2024; Xiao et al., 2025), or applying post-processing (Mazé & Ahmed, 2023) corrections af-

ter generation. While these approaches improve constraint adherence, they remain fundamentally limited. Training-time methods struggle to handle unseen constraints, guidance-based techniques offer only soft biases without guarantees, and post-processing can distort the distribution learned by the generative model. These issues highlight three key challenges that need to be jointly addressed: (1) guaranteeing state and action constraint satisfaction, (2) ensuring dynamic consistency so that trajectories are physically valid, and (3) handling novel test-time constraints without retraining.

To address these challenges, we propose **SAD-Flower** – a novel control-augmented flow matching framework designed to generate **S**afe, **A**dmissible, and **D**ynamically consistent trajectories. Inspired by guidance-based approaches, SAD-Flower introduces a virtual control input into the generation process. This control-theoretic interpretation of the sampling process enables a principled design of test-time guidance signals to ensure strong guarantees. The foundation of the design lies in a novel reformulation of state and action constraints into Control Barrier Function (CBF) conditions (Ames et al., 2017) for the generation process, while dynamic consistency is transformed into a Control Lyapunov Function (CLF) condition (Sontag, 1983). We exploit these conditions to design principled guidance signals via constrained optimal control problems, which can be solved efficiently using quadratic program solvers, such that SAD-Flower can handle novel constraints at test time without retraining. Unlike general constraint-projection approaches (Römer et al., 2025; Bouvier et al., 2025), which can significantly deviate from the original distribution learned by the generative model, we leverage prescribed-time control concepts (Song et al., 2017) to flexibly schedule constraint enforcement. This allows us to formally prove the safety, admissibility, and dynamic consistency of trajectories generated by SAD-Flower. These theoretical guarantees translate into empirical performance: across a range of domains, SAD-Flower consistently ensures constraint satisfaction while achieving competitive or superior task performance compared to existing generative planners. Notably, SAD-Flower remains robust even under increasingly stricter test-time constraints, validating its reliability in challenging deployment scenarios.

## 2 RELATED WORK

**Diffusion and Flow-Based Generative Models for Planning.** Recent advances in generative modeling, including diffusion models (Sohl-Dickstein et al., 2015; Ho et al., 2020; Song et al., 2021) and flow-matching (Lipman et al., 2023; Albergo & Vanden-Eijnden, 2023), have shown remarkable performance across various domains such as image generation (Dhariwal & Nichol, 2021; Du et al., 2020) and language modeling (Liu et al., 2023; Saharia et al., 2022). These generative approaches have also been successfully applied to data-driven planning, where the model learns to imitate expert behavior from datasets. For example, some works generate entire state-action trajectories directly using one (Janner et al., 2022; Zheng et al., 2023) or two separate models (Zhou et al., 2024), while others predict high-level trajectories and rely on a downstream controller to compute low-level actions (Chi et al., 2023; Ajay et al., 2023). However, these generative model planners operate without mechanisms to ensure that generated trajectories respect real-world constraints. In particular, they lack formal guarantees for satisfying state and action constraints, as well as dynamic consistency.

**Constraint-Aware Generative Model Planner.** To address the issue of constraint satisfaction, several recent works have explored constraint-aware planning based on generative models. Guidance-based methods (Dhariwal & Nichol, 2021; Yuan et al., 2023; Kondo et al., 2024; Ma et al., 2025; Carvalho et al., 2023) incorporate constraints by injecting gradients of auxiliary cost functions into the sampling process. While this encourages constraint satisfaction, it provides only a soft inductive bias without formal guarantees. Classifier-free guidance (Ho & Salimans, 2022) leverages information, such as constraint violation (Ajay et al., 2023) during training, enabling constraint-aware generation. However, these approaches require additional labeled data and have limited generalization to novel constraints. Similarly, DDAT (Bouvier et al., 2025) incorporates projection into the feasible set during training and inference but relies on strong assumptions, such as convexity of the constraint set. Post-processing approaches (Mazé & Ahmed, 2023; Giannone et al., 2023) attempt to enforce constraints by optimizing generated samples after denoising. However, these modifications are unaware of the learned data distribution and can produce samples that significantly drift from it.

**Control-Theoretic Enforcement in Generative Planning.** Several works have proposed control-theoretic techniques to enforce constraints. The works (Xiao et al., 2025; Botteghi et al., 2023; Dai et al., 2025) employ Control Barrier Functions (CBFs) (Ames et al., 2017) to enforce state con-

straints during the denoising process. However, these methods neglect action constraints and often produce trajectories that are not dynamically consistent and suffer from the local trap problem (Xiao et al., 2025). Control Lyapunov functions (CLFs) (Sontag, 1983) are leveraged together with CBFs in (Mizuta & Leung, 2024) to improve safety and stability, but formal guarantees are missing, and action constraints are not addressed. A constrained optimal control layer is integrated into the denoising process in (Römer et al., 2025) to enforce state and action constraints. Since it performs non-convex optimization throughout the entire sampling process, it exhibits a high computational cost and potentially steers samples prematurely before they reflect meaningful structure (Fan et al., 2025).

## 3 PROBLEM SETTING

We formally define the trajectory planning problem with safety, admissibility and dynamic consistency constraints as follows.

**System Model.** We consider a nonlinear dynamical system with state $\boldsymbol{s}(k) \in \mathbb{R}^n$ and action $\boldsymbol{a}(k) \in \mathbb{R}^m$ at time $k$, evolving as

$$\boldsymbol{s}(k+1) = \boldsymbol{f}(\boldsymbol{s}(k), \boldsymbol{a}(k)), \tag{1}$$

where $\boldsymbol{f}$ is the (possibly unknown) transition function. A trajectory is defined as a sequence of state-action pairs, $\boldsymbol{\tau} = \{(\boldsymbol{s}(0), \boldsymbol{a}(0)), \dots, (\boldsymbol{s}(H-1), \boldsymbol{a}(H-1))\}$. Given a dataset of expert trajectories $\mathcal{D} = \{\boldsymbol{\tau}^{(n)}\}_{n=1}^N$, our goal is to learn planning new trajectories that both imitate expert behavior and respect all deployment constraints.

**Constraints.** To ensure reliable real-world execution, every generated trajectory must, at every time step, satisfy: **(1) Safety:** the state remains within a safe set $\mathbb{S}$ (e.g., avoid collisions; eq. (**SC**)); **(2) Admissibility:** the action is within an admissible set $\mathbb{A}$ (e.g., satisfy torque or speed limits; eq. (**AC**)); and **(3) Dynamic Consistency:** the trajectory obeys the system dynamics (eq. (**DC**)). Neglecting any of these leads to unsafe, infeasible, or unrealizable plans: for example, trajectories may pass through obstacles (violating safety), demand unattainable actions (violating admissibility), or include state transitions that cannot be executed by the system (violating dynamics).

Formally, these requirements are posed on the distribution $p^{\boldsymbol{\theta}}(\boldsymbol{\tau})$ of the learned planner as follows:

$$\forall \boldsymbol{\tau} \sim p^{\boldsymbol{\theta}}(\boldsymbol{\tau}): \quad \boldsymbol{s}(k) \in \mathbb{S}, \quad \forall k = 0, \dots, H-1, \tag{SC}$$

$$\forall \boldsymbol{\tau} \sim p^{\boldsymbol{\theta}}(\boldsymbol{\tau}): \quad \boldsymbol{a}(k) \in \mathbb{A}, \quad \forall k = 0, \dots, H-1, \tag{AC}$$

$$\forall \boldsymbol{\tau} \sim p^{\boldsymbol{\theta}}(\boldsymbol{\tau}): \quad \boldsymbol{s}(k+1) = \boldsymbol{f}(\boldsymbol{s}(k), \boldsymbol{a}(k)), \quad \forall k = 0, \dots, H-1. \tag{DC}$$

**Objective.** Given expert data $\mathcal{D}$ and constraint sets $\mathbb{S}$, $\mathbb{A}$, our goal is to learn a generative model $p^{\boldsymbol{\theta}}(\boldsymbol{\tau})$ that (i) matches the expert trajectory distribution, and (ii) ensures all sampled trajectories satisfy eqs. (**SC**), (**AC**), and (**DC**). This setting motivates methods that can flexibly enforce constraints, even as requirements change at test time.

## 4 BACKGROUND: LEARNING TO PLAN WITH FLOW MATCHING

When a trajectory data set $\mathcal{D}$ of an expert planner is given, Flow Matching (FM) (Lipman et al., 2023; Zheng et al., 2023) is an effective technique to learn the distribution $p(\boldsymbol{\tau})$ of the data $\mathcal{D}$. In FM, the unknown distribution $p(\boldsymbol{\tau})$ is considered as the desired endpoint of a probability path $p_t^{\boldsymbol{\theta}}(\boldsymbol{\tau})$, $t \in [0, 1]$. The remainder of the probability path $p_t^{\boldsymbol{\theta}}(\boldsymbol{\tau})$ is characterized by a time-dependent vector field $\boldsymbol{v}_t^{\boldsymbol{\theta}} : [0, 1] \times \mathbb{R}^{(n+m)H} \to \mathbb{R}^{(n+m)H}$ parameterized by $\boldsymbol{\theta}$ that acts on samples $\boldsymbol{\tau}_0 \sim p_0(\boldsymbol{\tau})$ via the flow

$$\dot{\boldsymbol{\tau}}_t = \tfrac{d}{dt}\boldsymbol{\tau}_t = \boldsymbol{v}_t^{\boldsymbol{\theta}}(\boldsymbol{\tau}_t), \tag{2}$$

whereby the prior $p_0(\boldsymbol{\tau})$ is typically set to a Gaussian (Lipman et al., 2023). By prescribing a path from samples $\boldsymbol{\tau}_0$ of $p_0(\boldsymbol{\tau})$ to data trajectories $\boldsymbol{\tau}_1$ via a scheduled interpolation $\boldsymbol{\tau}_t = \alpha(t)\boldsymbol{\tau}_1 + \beta(t)\boldsymbol{\tau}_0$ with $\alpha$, $\beta$ such that $\alpha(0) = 0$, $\beta(1) = 0$ and $\alpha(t) + \beta(t) = 1$, the distribution learning problem is transformed into the supervised learning problem

$$\mathcal{L}_{\text{CFM}}(\boldsymbol{\theta}) = \mathbb{E}_{t \sim \mathcal{U}[0,1], \boldsymbol{\tau}_t \sim p_t^{\boldsymbol{\theta}}(\boldsymbol{\tau}), \boldsymbol{\tau}_1 \sim p(\boldsymbol{\tau})} ||\boldsymbol{v}_t^{\boldsymbol{\theta}}(\boldsymbol{\tau}_t) - \boldsymbol{v}_t(\boldsymbol{\tau}_0, \boldsymbol{\tau}_1)||_2^2, \tag{3}$$

where $\boldsymbol{v}_t(\boldsymbol{\tau}_0, \boldsymbol{\tau}_1) = \dot{\alpha}(t)\boldsymbol{\tau}_1 + \dot{\beta}(t)\boldsymbol{\tau}_0$ follows from the interpolation. Minimizing this cost function via stochastic gradient descent, $\boldsymbol{v}_t^{\boldsymbol{\theta}}(\boldsymbol{\tau}_t)$ can be efficiently trained using the trajectory data set $\mathcal{D}$.

Given an initial state $s_0$ and a trained vector field $v_t^\theta(\tau_t)$, we sample a random trajectory $\tau_0$ from the prior distribution $p_0$ and numerically solve the ordinary differential equation (ODE) in eq. (2) using $\tau_0$ as the initial condition to obtain $\tau_1$. Thereby, the trajectories effectively become samples $\tau_1 \sim p^\theta(\tau)$, where $p^\theta(\tau)$ is implicitly represented through $v_t^\theta(\tau_t)$. While such samples can be directly used in planning, they generally do not satisfy the safety, admissibility, and dynamic consistency constraints in eqs. (**SC**), (**AC**), and (**DC**).

**Remark 4.1.** *To allow plans with given initial states $s_0$, we only need to condition the initial distribution on $s(0) = s_0$ and ensure $\dot{\tau}_t^{s(0)} = 0$ for all $t \in [0, 1]$. For training, $s_0$ is chosen as the first state of trajectories in the data set, while arbitrary values can be set when sampling trajectories.*

## 5 Control Augmented Flow Matching for Constrained Planning

Despite the power of FM-based planners for trajectory generation, ensuring safety, admissibility, and dynamic consistency remains challenging, particularly under new test-time constraints. We address this with SAD-Flower, a control-augmented flow matching framework that provides formal guarantees for constraint satisfaction. In Section 5.1, we outline the control-theoretic intuition and its integration with FM. Section 5.2 details how constraint-aware quadratic programming augments sampling, and Section 5.3 presents theoretical guarantees of convergence and constraint satisfaction.

### 5.1 Control Augmentation for Safety, Admissibility and Dynamic Consistency

To ensure the satisfaction of safety (eq. (**SC**)), admissibility (eq. (**AC**)) and dynamic consistency (eq. (**DC**)), we extend the formulation in eq. (2) at test time to a controlled dynamical system

$$\dot{\tau}_t = v_t^\theta(\tau_t) + u_t, \tag{4}$$

where the vector field $v_t^\theta(\tau_t)$ represents the drift, while $u_t$ is a control input. By choosing $u_t = 0$, we recover standard flow matching as a special case of this formulation. Framing the problem in this way enables us to view the requirements in eqs. (**SC**), (**AC**), and (**DC**) as system properties, so that their satisfaction becomes a matter of control design with the following specifications.

**From State/Action Constraints to Barrier Specifications.** State and action constraints are set inclusion conditions, which require the controlled flow in eq. (4) to converge to and subsequently maintain constraint satisfaction. This behavior can be formalized via control barrier functions (CBFs) (Ames et al., 2017), whose level sets can encode the constraint sets $\mathbb{A}$ and $\mathbb{S}$. Thereby, we express state and action constraints as a condition on the growth of CBFs along the flow.

**From Dynamic Consistency to Lyapunov Specifications.** Dynamic consistency is an equality condition, whose violation needs to decay to $0$ along the flow in eq. (4). This property can be formalized using control Lyapunov functions (CLF) (Sontag, 1983) – energy-like, non-negative functions with a minimum of $0$ at the desired equilibrium. Hence, we formulate dynamic consistency as a condition on the decay of a CLF along the flow.

**Prescribed-time Specifications.** While FM simulates ODEs for time intervals $t \in [0, 1]$, Lyapunov and barrier specifications usually relate to asymptotic guarantees with $t \to \infty$. This discrepancy necessitates the scheduling of a sufficiently fast growth of CBFs and decrease of CLFs along the flow, which corresponds to a prescribed-time specification for control (Song et al., 2017).

Our approach – SAD-Flower – splits the numerical integration of eq. (4) into two phases as illustrated in Fig. 1. In the uncontrolled phase ($0 \leq t < T_0$), trajectories evolve under the learned FM vector field without intervention ($u_t = 0$) to preserve sample diversity (Fan et al., 2025). In the controlled phase ($T_0 \leq t \leq 1$), the control law $u_t$ satisfying CLF, CBF, and prescribed-time specifications is applied when integrating eq. (4). Thereby, the activation time $T_0 \in (0, 1)$ allows SAD-Flower to effectively balance generative flexibility with formal guarantees on safety (eq. (**SC**)), admissibility (eq. (**AC**)), and dynamic consistency (eq. (**DC**)).

### 5.2 Control Design using Control Lyapunov and Barrier Functions

Given the control specifications in Section 5.1, the actual control design problem remains. We first derive dedicated CBF and CLF constraints, which employ scheduling functions to ensure a sufficient growth/decrease rate. These constraints are exploited in an optimization-based control law.

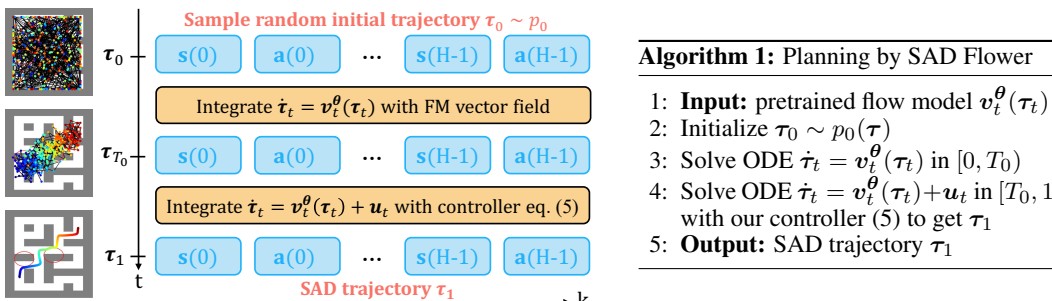

Figure 1: Overview of the trajectory generation using our proposed SAD-Flower.

**Control Barrier Constraints.** Since state/action constraints are specified separately for each time step, we design state CBFs $h_k^s(\boldsymbol{\tau}_t)$ and action CBFs $h_k^a(\boldsymbol{\tau}_t)$ for each time step $k = 0, \ldots, H-1$. Each CBF itself is defined as a signed distance function (SDF) (Park et al., 2019; Long et al., 2021), which measures the distance to the boundary of the set $\mathbb{S}$ and $\mathbb{A}$, respectively, and assigns a sign based on the inclusion in the constraint set.[1] Thus, these functions are only positive if states $\boldsymbol{s}(k)$ and actions $\boldsymbol{a}(k)$ are inside the sets $\mathbb{S}$ and $\mathbb{A}$, respectively. Non-negativity of the CBFs can be ensured by constraining the evolution of the CBF values along the flow, which results in the derivative condition

$$\dot{h}_k^s(\boldsymbol{\tau}_t) \geq -\varphi(t)h_k^s(\boldsymbol{\tau}_t), \quad \forall k = 1, \ldots, H-1, \tag{CBF-s}$$

$$\dot{h}_k^a(\boldsymbol{\tau}_t) \geq -\varphi(t)h_k^a(\boldsymbol{\tau}_t), \quad \forall k = 0, \ldots, H-1, \tag{CBF-a}$$

where $\dot{h}_k^{s/a}(\boldsymbol{\tau}_t) = \nabla^T h_k^{s/a}(\boldsymbol{\tau}_t) \left(\boldsymbol{v}_t^{\boldsymbol{\theta}}(\boldsymbol{\tau}_t) + \boldsymbol{u}_t\right)$ (Ames et al., 2017) and $\varphi(t)$ is a scheduling function that we will design later.

**Control Lyapunov Constraints.** For defining a suitable CLF, we sum up the squared consistency errors of a trajectory, i.e., $V(\boldsymbol{\tau}_t) = \frac{1}{2}\sum_{k=1}^{H-1}||\boldsymbol{s}(k) - \boldsymbol{f}(\boldsymbol{s}(k-1), \boldsymbol{a}(k-1))||^2$. This function is only 0 if the trajectory $\boldsymbol{\tau}_t$ is dynamically consistent, such that we constrain the evolution of its value along the flow to be negative via

$$\dot{V}(\boldsymbol{\tau}_t) \leq -\varphi(t)V(\boldsymbol{\tau}_t), \tag{CLF}$$

where $\dot{V}(\boldsymbol{\tau}_t) = \nabla^T V(\boldsymbol{\tau}_t)\left(\boldsymbol{v}_t^{\boldsymbol{\theta}}(\boldsymbol{\tau}_t) + \boldsymbol{u}_t\right)$ (Sontag, 1983) and $\varphi(t)$ is a scheduling function that we will design later. Computing $\nabla^T V(\boldsymbol{\tau}_t)$ requires access to $\boldsymbol{f}$. In some robotic domains this information is available through high fidelity simulators (Gaz et al., 2019; Howell et al., 2022; Acosta et al., 2022). If there exists no accurate physics-based simulator, e.g., for contact-rich manipulation tasks, a dynamics model can also be learned from the trajectory data. The learning errors directly correlate to the magnitude of dynamic consistency violations (Curi et al., 2020), such that practical consistency can still be achieved given sufficiently precise learned model.

**Prescribed-time Scheduling.** To guarantee that the CBFs are positive and the CLF is 0 at the terminal time $t = 1$ regardless of its state at $t = T_0$, we employ a scheduling function $\varphi(t) = \frac{c}{(1-t)^2}$ with some constant $c > 0$. Due to the steep growth of $\varphi$ for $t \to 1$, the constraints in eqs. (CBF-s), (CBF-a) and (CLF) become increasingly more restrictive. This ensures that positivity of CBFs and a vanishing CLF are ensured at some time $t < 1$ (Song et al., 2023).

**Constrained Minimum-Norm Optimal Control.** To ensure the satisfaction of CLF and CBF constraints, we formulate the minimum-norm optimal control problem

$$\boldsymbol{u}_t = \min_{\boldsymbol{u}} ||\boldsymbol{u}||^2 \qquad \text{s.t. eqs. (CBF-s), (CBF-a) and (CLF) hold,} \tag{5}$$

which ensures them by construction, while simultaneously minimizing the perturbation of the learned vector field $\boldsymbol{v}_t^{\boldsymbol{\theta}}(\boldsymbol{\tau}_t)$. Even though this optimization problem often consists of a large number of optimization variables and constraints, it can be solved comparatively efficiently since it is a quadratic program (QP). This renders the numerical integration of eq. (4) with the control law in eq. (5) computationally tractable when using dedicated QP solvers.

---

[1]For a formal definition of the CBFs $h_k^s(\boldsymbol{\tau}_t)$ and $h_k^a(\boldsymbol{\tau}_t)$, see Appendix A

### 5.3 THEORETICAL GUARANTEES FOR CONSTRAINT SATISFACTION AND CONSISTENCY.

Due to the strong theoretical foundations of CBFs and CLFs, strong guarantees for safety, admissibility, and dynamic consistency can be provided as shown in the following result.[2]

**Theorem 5.1.** *Assume that the QP in eq. (5) is feasible for all $t \in [T_0, 1]$. Then, the solution $\boldsymbol{\tau}_t$ of eq. (4) with control law $\boldsymbol{u}_t = \boldsymbol{0}$ for $t < T_0$ and $\boldsymbol{u}_t$ defined in eq. (5) for $t \geq T_0$ satisfies eqs. (SC), (AC), and (DC) at $t = 1$ for all initial conditions $\boldsymbol{\tau}_0$.*

This result shows that the validity of the safety, admissibility, and dynamic consistency guarantees essentially depends on the feasibility of the minimum-norm optimal control law in eq. (5), which is a fundamental requirement for a well-defined solution $\boldsymbol{\tau}_1$. While feasibility hinges on three technical conditions, it is not an issue in practice for the designed CLF and CBF constraints. Firstly, infeasibility often occurs only on sets of trajectories $\boldsymbol{\tau}$ with zero measure, such that it does not occur during the numerical integration of eq. (4). Secondly, many strategies for recovering feasibility exist, e.g., via slack variables (Boyd & Vandenberghe, 2004). If there exists a non-empty time interval including $t = 1$ after employing such a recovery strategy, it follows from Theorem 5.1 that eqs. (SC), (AC), and (DC) remain guaranteed. Hence, feasibility is usually not a practical concern.

## 6 EXPERIMENT

### 6.1 EXPERIMENT SETTING

We evaluate all methods in 4 different benchmark tasks: Maze2d, Hopper, and Walker2d from the D4RL problem set (Fu et al., 2020), and Kuka Block-Stacking (Janner et al., 2022).[3]

- **Maze2d** (Fu et al., 2020) is a navigation task where a point mass is moved from an initial state to a specified goal. Actions are artificially constrained to $[-0.1, 0.1]$. State constraints are defined for two novel obstacles, which do not block feasible paths and therefore still allow the task to be completed. Training data is generated by a navigation planner for the given maze. We evaluate on two maze configurations: Large and Umaze.
- **Hopper and Walker2d** (Fu et al., 2020) are locomotion tasks where a one-legged and a bipedal robot must move forward by jumping and walking, respectively. Actions are constrained to the range $[-1, 1]$. State constraints are imposed by requiring the robot's torso center to remain below a prescribed height (default: 1.6), creating a conflict with the objective of fast forward movement. We evaluate two datasets: demonstrations from a partially trained soft actor-critic policy (Haarnoja et al., 2018) (Medium) and a mixture of expert and partially trained policy data (Med-Expert).
- **Kuka Block-Stacking** (Janner et al., 2022) is a manipulation task where a 7-DOF robotic arm must stack a set of blocks. Unlike the other tasks, no action constraints or dynamic consistency requirements are enforced; only state constraints are applied, allowing us to study a variant of trajectory planning focused solely on state feasibility. These constraints ensure that self-collisions of the robot are avoided. Training data is generated using the PDDLStream planner (Garrett et al., 2020).

**Evaluation Metrics.** We evaluate the safety and admissibility violation of a trajectory via its maximal distance to the constraint sets $\mathbb{S}$ and $\mathbb{A}$, which we can effectively express through the CBFs as $-\min_k \min\{h_k^{s,a}(\boldsymbol{\tau}), 0\}$. Thus, a value of $0$ means constraint satisfaction, while positive values imply a violation. Dynamical consistency is measured using the Lyapunov function, such that large values indicate inconsistency. The performance of planned trajectories is measured by normalized total rewards for D4RL tasks and binary success rewards for stacking, as defined in Janner et al. (2022).

**Baselines:** We compare our proposed SAD-Flower against the following generative planners:

- **Diffuser** (Janner et al., 2022) generates trajectories using a diffusion model, without considering safety, admissibility, or dynamic consistency.
- **Truncation (Trunc)** (Brockman et al., 2016) enforces constraints by truncating the trajectory generated from the diffusion model.
- **Classifier Guidance (CG)** (Dhariwal & Nichol, 2021) augments diffusion-based trajectory generation with constraint-gradient guidance during sampling.

---

[2]A proof and extended discussion of the theorem's assumptions can be found in Appendix B.
[3]Details of the experimental setting are provided in the Appendix. D

- **Flow Matching (FM)** (Feng et al., 2025) trains a model with FM to generate trajectories, which is used as a baseline that does not incorporate safety, admissibility, or dynamic consistency.
- **SafeDiffuser (S-Diffuser)** (Xiao et al., 2025) generates trajectories via diffusion, while projecting states onto the constraint sets using a CBF at each sampling step.
- **Decision Diffuser (D-Diffuser)** (Ajay et al., 2023) trains a conditional diffusion model to generate state trajectories conditioned on task information and constraints, with the corresponding actions inferred from a learned inverse dynamics model.

## 6.2 Constrained-Planning Performance Across Benchmarks

As shown in Table 1, our method consistently satisfies safety and admissibility constraints while matching the planning performance of other methods across tasks. Although perfect dynamical consistency is not reached, the remaining violations are minor and mainly due to numerical integration of eq. (4). These results demonstrate the effectiveness of our control-theoretic mechanism, which guarantees constraint satisfaction at test time. SAD-Flower achieves this performance at the cost of merely a small computation time increase compared to existing methods (Appendix, Table 13).

In the navigation tasks of Maze2D, SAD-Flower achieves high rewards while maintaining complete constraint satisfaction. Since the task goals and imposed constraints are not in conflict, our method effectively balances planning performance and constraint adherence. Among baselines, SafeDiffuser employs a constraint-following mechanism during the diffusion sampling process to enforce state constraints, but it fails to guarantee admissibility. In contrast, Diffuser and FM achieve high rewards at the cost of violations in both safety and admissibility.

Table 1: Performance of the proposed SAD-Flower and baselines across navigation, locomotion, and manipulation tasks. The methods are compared on the maximum safety and admissibility constraint violations of planned trajectories, the magnitude of the dynamic consistency violation, and the model accuracy expressed through the reward. Truncate is not applicable in Maze2d (Umaze) due to more complex safety constraints, such that truncation becomes non-trivial (Xiao et al., 2025).

| Experiment | Metric | Diffuser | Trunc | CG | FM | S-Diffuser | D-Diffuser | Ours |
|---|---|---|---|---|---|---|---|---|
| Maze2d (Large) | safety | 0.43±0.39 | 0.10±0.25 | 0.27±0.37 | 0.37±0.39 | **0.00±0.00** | 0.83±0.24 | **0.00±0.00** |
| | admissib. | 0.89±0.01 | 0.88±0.07 | 0.87±0.02 | 0.90±0.00 | 0.91±0.02 | 0.90±0.03 | **0.00±0.00** |
| | dyn. consist. | 0.06±0.03 | 0.06±0.03 | 0.44±0.16 | 0.02±0.00 | 0.09±0.06 | 2.78±0.06 | **0.01 ± 0.01** |
| | reward | 1.40±0.26 | 1.39±0.26 | 0.40±0.33 | **1.43±0.20** | 1.20±0.06 | 0.38±0.18 | 1.42 ± 0.52 |
| Maze2d (Umaze) | safety | 0.04±0.20 | — | 0.51±0.33 | 0.11±0.22 | 0.87±3.77 | 0.05±0.15 | **0.00±0.00** |
| | admissib. | 0.90±0.01 | — | 0.88±0.03 | 0.01±0.00 | 0.90±0.02 | 0.89±0.02 | **0.00±0.00** |
| | dyn. consist. | 0.05±0.02 | — | 0.68±0.18 | **0.01±0.01** | 0.10±0.10 | 1.80±0.11 | **0.01±0.01** |
| | reward | 1.11±0.44 | — | 0.06±0.32 | 2.62±1.09 | 1.06±0.35 | 0.60±0.33 | **2.66±0.88** |
| Hopper (Med-Expert) | safety | 0.01±0.02 | 0.05±0.04 | 0.07±0.03 | 0.11±0.08 | 0.05±0.04 | 0.10±0.02 | **0.00±0.00** |
| | admissib. | 0.21±0.05 | 0.18±0.04 | 0.26±0.07 | 0.17±0.05 | 0.18±0.04 | **0.00±0.00** | **0.00±0.00** |
| | dyn. consist. | 0.38±0.03 | 0.41±0.04 | 0.79±0.10 | 0.23±0.02 | 0.36±0.06 | 0.16±0.01 | **0.01±0.01** |
| | reward | 1.06±0.18 | 0.50±0.12 | 0.73±0.22 | 1.02±0.20 | 0.53±0.19 | **1.12±0.01** | 0.93±0.23 |
| Hopper (Medium) | safety | 0.01±0.02 | 0.05±0.03 | **0.00±0.00** | 0.39±0.13 | 0.01±0.01 | 0.15±0.01 | **0.00±0.00** |
| | admissib. | 0.21±0.05 | 0.18±0.04 | 0.16±0.03 | 0.32±0.21 | 0.18±0.04 | **0.00±0.00** | **0.00±0.00** |
| | dyn. consist. | 0.46±0.01 | 0.47±0.01 | 0.95±0.02 | 0.42±0.39 | 0.47±0.01 | 0.18±0.01 | **0.01±0.01** |
| | reward | 0.44±0.05 | 0.45±0.06 | 0.39±0.03 | **0.49±0.05** | 0.45±0.06 | 0.48±0.08 | 0.34±0.03 |
| Walker2d (Med-Expert) | safety | 0.06±0.04 | 0.06±0.05 | 0.02±0.03 | 0.40±0.11 | 0.09±0.07 | 0.04±0.04 | **0.00±0.00** |
| | admissib. | 0.67±0.18 | 0.62±0.14 | 0.72±0.18 | 0.15±0.02 | 0.58±0.20 | **0.00±0.00** | **0.00±0.00** |
| | dyn. consist. | 0.71±0.05 | 0.72±0.05 | 0.83±0.91 | 0.44±0.01 | 0.79±0.03 | 0.69±0.05 | **0.04±0.04** |
| | reward | 1.06±0.23 | 0.56±0.29 | 0.39±0.19 | **1.07±0.01** | 0.59±0.21 | 0.95±0.24 | 0.89±0.32 |
| Walker2d (Medium) | safety | 0.03±0.03 | 0.02±0.01 | 0.02±0.02 | 0.21±0.15 | 0.02±0.02 | 0.09±0.04 | **0.00±0.00** |
| | admissib. | 0.56±0.10 | 0.44±0.19 | 0.54±0.14 | 0.48±0.06 | 0.52±0.12 | **0.00±0.00** | **0.00±0.00** |
| | dyn. consist. | 0.68±0.08 | 0.65±0.08 | 0.72±0.38 | 0.40±0.06 | 0.64±0.07 | 1.52±0.05 | **0.07±0.15** |
| | reward | 0.57±0.26 | 0.50±0.26 | 0.55±0.28 | 0.73±0.15 | 0.49±0.23 | **0.76 ± 0.16** | 0.42±0.23 |
| KUKA Block Stacking | safety | 0.23±0.09 | **0.00±0.00** | 0.22±0.09 | 0.02±0.04 | **0.00±0.00** | 0.14±0.13 | **0.00±0.00** |
| | reward | 0.46±0.23 | 0.45±0.21 | 0.45±0.23 | 0.44±0.20 | 0.49±0.23 | **0.55±0.26** | 0.45±0.21 |

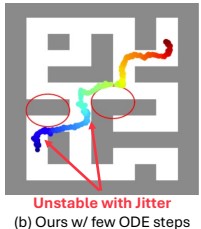 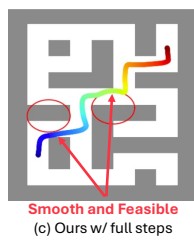

Figure 2: (a) Without enforcing dynamic consistency, applying state and action constraints can result in significant outliers, known as the local trap problem (Xiao et al., 2025). (b) Our method satisfies constraints, but using too few numerical integration steps for the ODE introduces jitter in the trajectory. (c) With sufficient integration steps, our method produces dynamically consistent trajectories while satisfying unseen constraints (red ellipses).

In locomotion tasks, where dynamics are more complex, SAD-Flower is the only method that satisfies all constraints while maintaining strong dynamical consistency. Its slightly lower rewards compared to baselines stem from conflicts between the imposed height constraint and the jumping or walking behaviors required for high returns, making safety violations directly correlated with the rewards that reflect planning performance. For instance, CG achieves planning performance comparable to SAD-Flower in Hopper (Med-Expert) when both methods avoid safety violations. While some methods, such as Decision Diffuser, consistently ensure admissibility, all struggle with dynamical consistency due to the complexity of the robot dynamics.

In the Kuka Block-Stacking task, which excludes admissibility and dynamical consistency requirements, SAD-Flower leverages the simplified setting to guarantee safety while achieving rewards competitive with other safety-enforcing baselines. This highlights both the effectiveness and flexibility of our proposed approach.

## 6.3 WHY DOES SAD-FLOWER WORK EFFECTIVELY?

We analyze three key properties of SAD-Flower that contribute to its effectiveness.

**Dynamic consistency prevents local traps.** Safety and admissibility can be enforced by projecting states or actions back into the constraint sets, but this can introduce sharp discontinuities in the trajectory, as illustrated in Fig. 2(a) for SafeDiffuser in Maze2D. This phenomenon, known as the local trap problem (Xiao et al., 2025), arises because constraints are treated independently at each trajectory step. In contrast, SAD-Flower enforces dynamic consistency by coupling consecutive states and actions through the CLF, ensuring coherent evolution during integration of the flow. This coupling prevents misaligned guidance and eliminates the risk of local traps, as demonstrated in Fig. 2(c).

**Delayed control activation avoids premature interventions.** At the beginning of the flow sampling process, sample distributions are relatively unstructured, and applying control signals too early can push trajectories away from behaviors captured by the learned generative planner. Perturbations introduced in this phase are propagated throughout the flow in eq. (4), potentially degrading performance. SAD-Flower mitigates this issue by activating guidance only after a prescribed flow time $T_0$. As shown in Fig. 3 (left) for Maze2D, even a small $T_0$ suffices to achieve high rewards, while a larger $T_0$ can further improve performance.

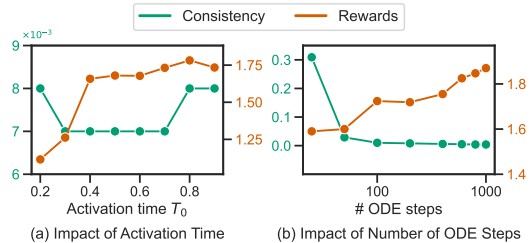

Figure 3: Effect of increasing the number of ODE steps for solving eq. (4) and the activation time $T_0$ of controller eq. (5) on the consistency and performance of trajectories.

Dynamic consistency violations increase only marginally with larger $T_0$. Safety and admissibility are satisfied across all $T_0$ values in our experiments (not shown due to space). These results highlight the effectiveness of our control-theoretic formulation, which allows delayed activation of CLF and CBF guidance without compromising constraints or dynamic consistency.

**QP vs. non-convex optimization.** To assess the benefits of our QP-based formulation, we compare SAD-Flower with DPCC (Römer et al., 2025), a recent method

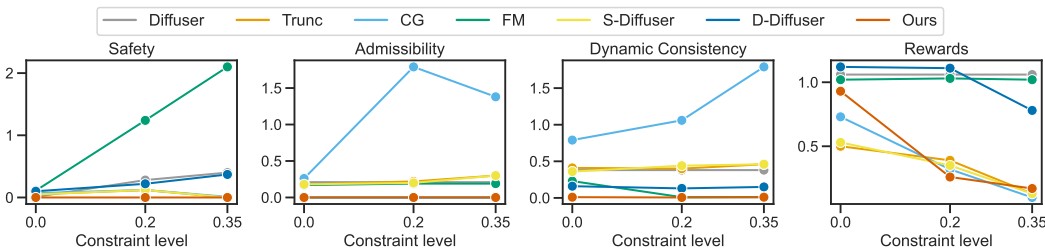

Figure 4: Performance of the proposed SAD-Flower and baselines depending on level of constraint tightening for the Hopper (Med-Expert) benchmark.

that also integrates optimal control into generative planning to enforce safety, admissibility, and dynamic consistency. DPCC applies control from the earliest stages and relies on full-system non-convex optimization at every sampling step.

As shown in Table 2 for Hopper (Med-Expert), both methods achieve comparable levels of admissibility and dynamic consistency, but DPCC incurs much higher computation per sampling step and lower rewards. This highlights the efficiency of SAD-Flower gained without any loss of constraint satisfaction by enabling the usage of QP solvers compared to general non-convex optimization methods. Note though that the enforcement of constraints does still not come for free with QPs since the integration steps of the uncontrolled ODE merely takes 0.01s on average.

Table 2: Benefits of QPs in SAD-flower compared to general non-convex optimization of DPCC (Römer et al., 2025).

| Experiment | Metric | DPCC | Ours |
|---|---|---|---|
| Hopper (Med-Expert) | safety | 0.01±0.03 | **0.00±0.00** |
| | admissib. | **0.00±0.00** | **0.00±0.00** |
| | dyn. consist. | **0.01±0.01** | **0.01±0.01** |
| | reward | 0.61±0.07 | **0.93±0.23** |
| | comp. time [s] | 4.24±1.64 | **0.06±0.00** |

### 6.4 How reliable is SAD-Flower?

To highlight the reliability of SAD-Flower, we demonstrate its behavior when approaching the extremes in terms of approximations for implementation and problem difficulty.

**Reducing integration accuracy.** We first illustrate the reliability of SAD-Flower when the accuracy of integrating the controlled ODE in eq. (4) is small, which we measure through the number of ODE discretization steps. As illustrated on the right of Fig. 3, increasing the number of ODE steps improves dynamic consistency since the controller in eq. (5) has more opportunities to intervene. Even with as few as 25 ODE steps, the inconsistency is relatively small and corresponds only to minor jittering behavior in the trajectories as illustrated in Fig. 2. With $\approx 100$ ODE steps, dynamic consistency is practically achieved. Safety and admissibility are ensured for the whole range of considered ODE steps (not depicted). In contrast, the rewards continue to grow for finer discretizations, which cause higher computational complexity. Thus, SAD-Flower allows to trade-off performance and complexity, while simultaneously ensuring constraint satisfaction and dynamic consistency.

**Handling stricter test-time constraints.** To evaluate the robustness of our method under unseen conditions, we test it with increasingly restrictive constraints. For this, we reduce the allowed torso height by 0.2 and 0.35 in the Hopper environment. As shown in Fig. 4, SAD-Flower maintains perfect constraint satisfaction even under significantly stricter settings. Although the rewards achieved by SAD-Flower decrease with tighter constraints, this effect is caused by the conflict of constraints and objective. Thus, other methods enforcing state constraints, e.g., SafeDiffuser, also exhibit this behavior. Notably, the rewards of SAD-Flower are generally at least on par with these methods, even though the baselines exhibit significant admissibility and dynamic consistency violations. This clearly demonstrates the strong robustness of SAD-Flower in handling unseen test-time constraints.

**Scalability in high-dimensional tasks.** To evaluate the scalability of our approach in complex, contact-rich manipulation settings, we additionally consider the D4RL Adroit Relocate task, which involves a 39-dimensional state space and a 30-dimensional action space. This dexterous grasping environment poses significant challenges due to the need for precise contact interactions and high-dimensional control. We define the state constraint via

Table 3: Performance of SAD-Flower and FM in a dexterous grasping scenario.

| Experiment | Metric | FM | Ours |
|---|---|---|---|
| Adroit-Hand (Relocate) | safety | 0.15±0.21 | **0.00±0.00** |
| | admissib. | 0.62±0.19 | **0.00±0.00** |
| | dyn. consist. | 0.07±0.04 | **0.06±0.09** |
| | reward | **1.07±0.08** | 1.05±0.05 |

the arm's position to avoid collisions and the action constraint through actuator limits. As shown in Table 3, standard flow matching achieves moderate task success but suffers from frequent violations of safety and admissibility constraints due to unconstrained sampling. In contrast, SAD-Flower consistently satisfies both state and action constraints, demonstrating its ability to scale to high-dimensional robotic systems while maintaining constraint enforcement. This result underscores the robustness and generality of our control-augmented framework in realistic manipulation tasks.

**Sensitivity analysis of constraint-enforcement strength.** We conduct an ablation study on the LargeMaze task to assess the sensitivity of SAD-Flower to the hyperparameter $c$, which modulates the strength of constraint enforcement. While the main experiments use $c = 0.5$, we vary it from 1.0 to 0.2. As shown in Table 4, SAD-Flower maintains perfect safety and admissibility across all tested values. Violations of dynamic consistency remain negligible and stable, while task performance varies moderately. These results demonstrate that SAD-Flower achieves constraint satisfaction without requiring precise hyperparameter tuning, highlighting its practical reliability.

**Robustness under imperfect dynamics models.** To evaluate the robustness of our method under model approximation errors, we simulate degraded dynamics by training the forward model on progressively smaller subsets of the Hopper-Medium dataset. As shown in Table 5, SAD-Flower maintains full constraint satisfaction even with only 10% of the original data, and dynamic consistency remains stable across these settings. Degradation becomes apparent only when the dataset is reduced to 0.01%, at which point safety violations emerge. These results demonstrate that SAD-Flower is tolerant to moderately inaccurate dynamics models and that training a sufficiently accurate model is feasible in practice.

Table 4: Sensitivity of SAD-Flower to the hyperparameter $c$.

| Experiment | Metric | 1.0 | 0.8 | 0.6 | 0.4 | 0.2 |
|---|---|---|---|---|---|---|
| Maze2d (Large) | safety | **0.00±0.00** | **0.00±0.00** | **0.00±0.00** | **0.00±0.00** | **0.00±0.00** |
| | admissib. | **0.00±0.00** | **0.00±0.00** | **0.00±0.00** | **0.00±0.00** | **0.00±0.00** |
| | dyn. consist. | **0.01±0.01** | **0.01±0.01** | **0.01±0.01** | **0.01±0.01** | 0.02 ± 0.01 |
| | reward | 1.38±0.59 | **1.54±0.59** | 1.46±0.52 | 1.47±0.53 | 1.29 ± 0.73 |

Table 5: Effect of training dataset size on dynamic consistency and constraint satisfaction.

| Experiment | Metric | 90% | 70% | 50% | 30% | 10% | 0.01% |
|---|---|---|---|---|---|---|---|
| Hopper (Medium) | safety | 0.00±0.00 | 0.00±0.00 | 0.00±0.00 | 0.00±0.00 | 0.00±0.00 | 0.02±0.09 |
| | admissib. | 0.00±0.00 | 0.00±0.00 | 0.00±0.00 | 0.00±0.00 | 0.00±0.00 | 0.00±0.00 |
| | dyn. consist. | 0.01±0.02 | 0.01±0.01 | 0.01±0.01 | 0.01±0.02 | 0.01±0.01 | 0.04±0.02 |
| | reward | 0.35±0.01 | 0.33±0.04 | 0.35±0.07 | 0.39±0.07 | 0.38±0.03 | 0.36±0.03 |

## 7  CONCLUSION

We presented SAD-Flower, a control-augmented flow matching framework that ensures safe, admissible, and dynamically consistent trajectory planning. By reformulating flow matching as a controllable dynamical system with a virtual control input, our method leverages Control Barrier Function and Control Lyapunov Function conditions scheduled using prescribed-time control principles to enforce constraints at test time without retraining. Experiments across navigation, locomotion, and manipulation tasks show that SAD-Flower achieves perfect constraint satisfaction, avoids local traps, and maintains competitive task performance. These results establish it as a practical, theoretically grounded solution for real-world deployment. Looking ahead, extending SAD-Flower to stochastic dynamics, complex constraint geometries, and online replanning offers promising directions for safe and reliable generative trajectory planning.

REPRODUCIBILITY STATEMENT

**Code.** Our method is implemented based on the publicly available codebase from (Feng et al., 2025) at `https://github.com/AI4Science-WestlakeU/flow_guidance`. The code

for our method will be released upon acceptance. Implementation and experimental details are provided in section 6 of the main text, with further information in the Appendix.

**Theory.** Our theoretical claims in Section 5.3 are supported by complete proofs in Appendix B.

**Datasets.** We evaluate our method on a variety of public datasets across locomotion, navigation, and manipulation tasks. Full descriptions of the datasets can be found in Section 6 and Appendix D.

**Compute.** All experiments were conducted using NVIDIA Tesla P100 GPUs. Additional information on hardware and computational resources is provided in Appendix G.

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

APPENDIX

## Table of Contents

## A   SIGNED DISTANCE FUNCTIONS

Given sets $\mathbb{S}$ and $\mathbb{A}$, the signed distance functions are defined as the minimal distance to a point on the boundary $\partial\mathbb{S}$ and $\partial\mathbb{A}$ of the sets, with the sign indicating if the state/action along the trajectory is in the corresponding sets. This formally leads to the following functions

$$h_k^s(\boldsymbol{\tau}) = \begin{cases} \min_{\boldsymbol{s}\in\partial\mathbb{S}}||\boldsymbol{s}(k)-\boldsymbol{s}|| & \text{if } \boldsymbol{s}\in\mathbb{S} \\ -\min_{\boldsymbol{s}\in\partial\mathbb{S}}||\boldsymbol{s}(k)-\boldsymbol{s}|| & \text{else} \end{cases} \qquad h_k^a(\boldsymbol{\tau}) = \begin{cases} \min_{\boldsymbol{a}\in\partial\mathbb{A}}||\boldsymbol{a}(k)-\boldsymbol{a}|| & \text{if } \boldsymbol{a}\in\mathbb{A} \\ -\min_{\boldsymbol{a}\in\partial\mathbb{A}}||\boldsymbol{a}(k)-\boldsymbol{a}|| & \text{else.} \end{cases} \tag{6}$$

## B   PROOFS OF THEORETICAL RESULTS

### B.1   FEASIBILITY OF CBF CONSTRAINTS

**Theorem B.1.** *Assume that $\varphi(t) = \frac{c}{(t-1)^2}$ with constant $c \in \mathbb{R}_+$. If $h_k^s$ and $h_k^a$ are differentiable, then, for each $\boldsymbol{\tau}$, there exists a $\boldsymbol{u}$ jointly satisfying eq. (CBF-s) and eq. (CBF-a).*

*Proof.* Since the constraints defined in eqs. (CBF-s) and (CBF-a) concern independent variables, individual feasibility of each constraint implies joint feasibility of all of them. Due to the assumed differentiability of $h_k^{s,a}$ and the definition of SDFs implying $||\nabla h_k^{s,a}(\boldsymbol{\tau})|| = 1$, $\boldsymbol{u}$ can always be chosen such $\nabla^T h_k^{s,a}(\boldsymbol{\tau})\boldsymbol{u}$ takes an arbitrary value. Thus, the constraints defined in eqs. (CBF-s) and (CBF-a) are always feasible. $\qquad\square$

The assumption on the differentiability of $h_k^s$ and $h_k^a$ is required as the signed distance functions (SDF) eq. (6) are generally not smooth. However, they are differentiable almost everywhere for sets with smooth boundary under weak assumptions (Gilbarg & Trudinger, 1977). Non-differentiable set boundaries can be overcome by increasing the number of CBF constraints via a decomposition of $\bar{\mathbb{S}}$ and $\bar{\mathbb{A}}$ into suitable subsets $\bar{\mathbb{S}}_i$ and $\bar{\mathbb{A}}_i$, such that the SDFs can be computed with respect to the subsets instead. For example, this approach immediately yields linear functions of the form $h_{k,i}^a(\boldsymbol{\tau}) = \bar{a} \pm \boldsymbol{a}_i(k)$ for individual elements $\boldsymbol{a}_i(k)$ of actions when defining suitable subsets $\bar{\mathbb{A}}_i$ for the the commonly employed box constraints $\|\boldsymbol{a}_i(k)\|_\infty \leq \bar{a}$ with some constant $a \in \mathbb{R}_+$. Thus the differentiability assumption in Theorem B.1 is rather technical and often not relevant in practice for the usage of SDFs in CBF constraints (Long et al., 2021).

## B.2 FEASIBILITY OF CLF CONSTRAINTS

**Theorem B.2.** *Assume that* $\varphi(t) = \frac{c}{(t-1)^2}$ *with constant* $c \in \mathbb{R}_+$. *If* $\{(\boldsymbol{s}, \boldsymbol{a}) : \det(\frac{\partial}{\partial \boldsymbol{s}}\boldsymbol{f}(\boldsymbol{s}, \boldsymbol{a})) = \boldsymbol{0} \wedge \det(\frac{\partial}{\partial \boldsymbol{a}}\boldsymbol{f}(\boldsymbol{s}, \boldsymbol{a})) = \boldsymbol{0}\} = \emptyset$, *then, for each* $\boldsymbol{\tau}$, *there exists a* $\boldsymbol{u}$ *satisfying eq. (CLF).*

*Proof.* Let

$$\boldsymbol{\ell}_k(\tau) = \left(\boldsymbol{\tau}^{\boldsymbol{s}(k+1)} - \boldsymbol{f}(\boldsymbol{\tau}^{\boldsymbol{s}(k)}, \boldsymbol{\tau}^{\boldsymbol{a}(k)})\right). \tag{7}$$

Then, the gradient of $V$ is given by

$$\nabla V(\boldsymbol{\tau}) = \begin{bmatrix} -\frac{\partial \boldsymbol{f}(\boldsymbol{\tau}^{\boldsymbol{s}(0)}, \boldsymbol{\tau}^{\boldsymbol{a}(0)})}{\partial \boldsymbol{\tau}^{\boldsymbol{s}(0)}} \boldsymbol{\ell}_0(\boldsymbol{\tau}) \\ -\frac{\partial \boldsymbol{f}(\boldsymbol{\tau}^{\boldsymbol{s}(0)}, \boldsymbol{\tau}^{\boldsymbol{a}(0)})}{\partial \boldsymbol{\tau}^{\boldsymbol{a}(0)}} \boldsymbol{\ell}_0(\boldsymbol{\tau}) \\ \boldsymbol{\ell}_0(\boldsymbol{\tau}) - \frac{\partial \boldsymbol{f}(\boldsymbol{\tau}^{\boldsymbol{s}(1)}, \boldsymbol{\tau}^{\boldsymbol{a}(1)})}{\partial \boldsymbol{\tau}^{\boldsymbol{s}(1)}} \boldsymbol{\ell}_1(\boldsymbol{\tau}) \\ -\frac{\partial \boldsymbol{f}(\boldsymbol{\tau}^{\boldsymbol{s}(1)}, \boldsymbol{\tau}^{\boldsymbol{a}(1)})}{\partial \boldsymbol{\tau}^{\boldsymbol{a}(1)}} \boldsymbol{\ell}_1(\boldsymbol{\tau}) \\ \vdots \\ \boldsymbol{\ell}_{H-2}(\boldsymbol{\tau}) - \frac{\partial \boldsymbol{f}(\boldsymbol{\tau}^{\boldsymbol{s}(H-1)}, \boldsymbol{\tau}^{\boldsymbol{a}(H-1)})}{\partial \boldsymbol{\tau}^{\boldsymbol{s}(H-1)}} \boldsymbol{\ell}_{H-1}(\boldsymbol{\tau}) \\ -\frac{\partial \boldsymbol{f}(\boldsymbol{\tau}^{\boldsymbol{s}(H-1)}, \boldsymbol{\tau}^{\boldsymbol{a}(H-1)})}{\partial \boldsymbol{\tau}^{\boldsymbol{a}(H-1)}} \boldsymbol{\ell}_{H-1}(\boldsymbol{\tau}) \\ \boldsymbol{\ell}_{H-1}(\boldsymbol{\tau}) \\ \boldsymbol{0} \end{bmatrix}. \tag{8}$$

In the following, we will show by contradiction that $\nabla V(\boldsymbol{\tau}) = \boldsymbol{0}$ if and only if $\boldsymbol{\ell}_k(\boldsymbol{\tau}) = \boldsymbol{0}$ for all $k = 0, \ldots, H-1$. For this purpose, assume that $\nabla V(\boldsymbol{\tau}) = \boldsymbol{0}$ and $\boldsymbol{\ell}_i \neq \boldsymbol{0}$ for some $i = 1, \ldots, H-1$. If $\mathrm{rank}(\frac{\partial \boldsymbol{f}(\boldsymbol{\tau}^{\boldsymbol{s}(i)}, \boldsymbol{\tau}^{\boldsymbol{a}(i)})}{\partial \boldsymbol{\tau}^{\boldsymbol{a}(i)}}) \neq 0$, $\nabla V(\boldsymbol{\tau}) \neq \boldsymbol{0}$ is trivially contradicted. If $\mathrm{rank}(\frac{\partial \boldsymbol{f}(\boldsymbol{\tau}^{\boldsymbol{s}(i)}, \boldsymbol{\tau}^{\boldsymbol{a}(i)})}{\partial \boldsymbol{\tau}^{\boldsymbol{s}(i)}}) \neq 0$, $\nabla V(\boldsymbol{\tau}) = \boldsymbol{0}$ requires $\boldsymbol{\ell}_{i-1} \neq \boldsymbol{0}$. Hence, we can consider the same two cases as for $\boldsymbol{\ell}_i \neq \boldsymbol{0}$. By repeating this procedure and always considering the case $\mathrm{rank}(\frac{\partial \boldsymbol{f}(\boldsymbol{\tau}^{\boldsymbol{s}(k)}, \boldsymbol{\tau}^{\boldsymbol{a}(k)})}{\partial \boldsymbol{\tau}^{\boldsymbol{s}(k)}}) \neq 0$, we eventually end up with the condition

$$\frac{\partial \boldsymbol{f}(\boldsymbol{\tau}^{\boldsymbol{s}(0)}, \boldsymbol{\tau}^{\boldsymbol{a}(0)})}{\partial \boldsymbol{\tau}^{\boldsymbol{s}(0)}} \boldsymbol{\ell}_0(\boldsymbol{\tau}) = \boldsymbol{0} \qquad \wedge \qquad \mathrm{rank}(\frac{\partial \boldsymbol{f}(\boldsymbol{\tau}^{\boldsymbol{s}(0)}, \boldsymbol{\tau}^{\boldsymbol{a}(0)})}{\partial \boldsymbol{\tau}^{\boldsymbol{a}(0)}}) \neq 0 \qquad \wedge \qquad \boldsymbol{\ell}_0 \neq \boldsymbol{0}, \tag{9}$$

which cannot be satisfied. Thus, $\nabla V(\boldsymbol{\tau}) \neq \boldsymbol{0}$ holds if $\boldsymbol{\ell}_k(\boldsymbol{\tau}) \neq \boldsymbol{0}$ for some $k$. Consequently, there always exists a $\boldsymbol{u}$ such that eq. (CLF) is satisfied rendering the constraint feasible. $\square$

To ensure the feasibility of CLF constraints, we again need one technical assumption: Through infinitesimal changes of states or actions, the dynamics $\boldsymbol{f}$ can be changed in arbitrary directions. If this property is not satisfied, $\nabla V(\boldsymbol{\tau})$ does not necessarily provide information about directions for reducing the violation of eq. (**DC**). Note that matrices with rank deficiency have zero measure among all matrices, such that infeasibilities related to a violation of the rank condition in Theorem B.2 occur only at isolated states (except for special cases of dynamics, e.g., piecewise constant dynamics). Thus, the rank condition is usually satisfied almost everywhere for many relevant systems, which is sufficient for the feasibility of the CLF constraint in eq. (CLF) in practice.

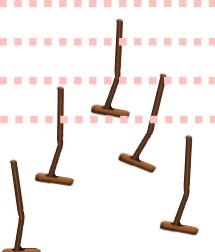

Figure 5: Performance under progressively tightened constraints in the locomotion task. As the allowed torso height decreases, the admissibility constraint becomes stricter, creating a conflict with the task objective of jumping forward.

### B.3 PROOF OF THEOREM 5.1

*Proof of Theorem 5.1.* Due to the assumed feasibility of constraints, eq. (4) controlled by eq. (5) satisfies eq. (CBF-s), eq. (CBF-a), and eq. (CLF), i.e., prescribed-time CBF and CLF conditions are satisfied. As $h_k^{s,a}$ is positive inside $\mathbb{S}/\mathbb{A}$, negative outside, and zero on the boundaries of these sets, it corresponds to a prescribed-time control barrier function (Huang et al., 2024). Therefore, it immediately follows from (Huang et al., 2024, Theorem 1) that $h_k^{s,a}(\boldsymbol{\tau}_1) > 0$ at $t = 1$, which implies satisfaction of safety (eq. (**SC**)) and admissibility (eq. (**AC**)) by construction. Since $V$ is positive definite, it corresponds to a prescribed-time control Lyapunov function (Song et al., 2019). Hence, it immediately follows from (Song et al., 2017, Theorem 2) that $\boldsymbol{V}(\boldsymbol{\tau}_1) = 0$, which implies satisfaction of eq. (**DC**) by construction. □

While the two constraints in eq. (CBF-s) and eq. (CBF-a) concern independent variables, the addition of constraint eq. (CLF) introduces a coupling between the constraints. This coupling can cause infeasibility at some trajectories $\boldsymbol{\tau}$ in general, but technical conditions exist that exclude them (Wang et al., 2024). Moreover, these infeasibilities are not an issue from a practical perspective since they often affect isolated $\boldsymbol{\tau}$ or small subsets of trajectories usually not occurring while numerically integrating eq. (4).

## C ADDITIONAL EXPERIMENT DETAILS AND RESULTS

**Robustness to Stricter Test-Time Constraints.** We evaluate the robustness of SAD-Flower by tightening the test-time constraint on torso height (Fig. 5) in the Hopper environment, decreasing the upper bound from 1.6 to 1.4 and 1.25 settings not encountered during training. As shown in Table 6, our method maintains perfect satisfaction of both safety and admissibility constraints across all levels of difficulty, despite the increasingly restrictive conditions. This demonstrates SAD-Flower's strong generalization to stricter, unseen constraints at test time. While task rewards decrease modestly—as expected due to the heightened challenge—our method consistently preserves formal guarantees of constraint satisfaction, validating the effectiveness of prescribed-time control as a flexible enforcement mechanism even under distributional shifts.

**Comparision with DPCC (Römer et al., 2025).** To provide a more comprehensive comparison with optimization-based generative planning, we evaluate SAD-Flower against three DPCC variants (Römer et al., 2025): DPCC-T, which selects samples with minimal deviation from the previous timestep; DPCC-C, which minimizes cumulative projection cost; and DPCC-R, which selects samples randomly after projection. All DPCC variants leverage full-system nonlinear optimization at each denoising step to enforce constraints. As shown in Table 7, while all DPCC variants succeed in satisfying admissibility and achieving dynamic consistency, they exhibit consistently higher safety violations (up to 0.02) even when we apply the constraint tightening technique (using height=1.5 under the constraint with height=1.6) and incur significantly greater computational cost—up to 25× slower than SAD-Flower. These issues likely stem from performing optimization over early-stage noisy samples, which may be difficult or even infeasible to correct without distorting future structure. Furthermore, the lower planning reward across all DPCC variants (0.61–0.65 vs. 0.93) suggest that

Table 6: Performance of the proposed SAD-Flower and baselines depending on constraint tightness. Constraints for the Hopper are tightened via lower admissible heights.

| Experiment | Metric | Diffuser | Truncate | CG | FM | S-Diffuser | D-Diffuser | Ours |
|---|---|---|---|---|---|---|---|---|
| Hopper (height=1.6) | safety | 0.01±0.02 | 0.05±0.04 | 0.07±0.03 | 0.11±0.08 | 0.05±0.04 | 0.10±0.02 | **0.00±0.00** |
| | admissib. | 0.21±0.05 | 0.18±0.04 | 0.26±0.07 | 0.17±0.05 | 0.18±0.04 | **0.00±0.00** | **0.00±0.00** |
| | dyn. consist. | 0.38±0.03 | 0.41±0.04 | 0.79±0.10 | 0.23±0.02 | 0.36±0.06 | 0.16±0.01 | **0.01±0.01** |
| | reward | 1.06±0.18 | 0.50±0.12 | 0.73±0.22 | 1.02±0.20 | 0.53±0.19 | **1.12±0.01** | 0.93±0.23 |
| Hopper (height=1.4) | safety | 0.28±0.08 | 0.12±0.09 | 0.04±0.03 | 1.24±0.07 | 0.12±0.09 | 0.22±0.03 | **0.00±0.00** |
| | admissib. | 0.21±0.05 | 0.22±0.09 | 1.79±0.33 | 0.19±0.10 | 0.20±0.08 | **0.00±0.00** | **0.00±0.00** |
| | dyn. consist. | 0.38±0.03 | 0.40±0.08 | 1.06±0.03 | 0.01±0.01 | 0.44±0.06 | 0.13±0.01 | **0.01±0.01** |
| | reward | 1.06±0.18 | 0.39±0.17 | 0.32±0.15 | 1.03±0.17 | 0.35±0.10 | **1.11±0.02** | 0.26±0.02 |
| Hopper (height=1.25) | safety | 0.40±0.04 | **0.00±0.00** | 0.01±0.01 | 2.10±0.07 | **0.00±0.00** | 0.37±0.02 | **0.00±0.00** |
| | admissib. | 0.21±0.05 | 0.30±0.05 | 1.38±0.27 | 0.05±0.10 | 0.30±0.05 | **0.00±0.00** | **0.00±0.00** |
| | dyn. consist. | 0.38±0.03 | 0.46±0.21 | 1.79±0.08 | **0.01±0.01** | 0.46±0.21 | 0.15±0.01 | **0.01±0.01** |
| | reward | 1.06±0.18 | 0.13±0.02 | 0.10±0.05 | **1.02±0.16** | 0.13±0.02 | 0.78±0.02 | 0.17±0.00 |

Table 7: The performance of the proposed SAD-Flower with QP and DPCC (Römer et al., 2025) with nonlinear optimization. Constraint for the Hopper is the same as the admissible heights in Table 1. Three DPCCs are proposed in their work. DPCC-T (Temporal consistency) selects the trajectory that deviates the least from the previous timestep, DPCC-C (Cumulative projection cost) selects the trajectory that has been modified the least by the projection operation, and DPCC-R (Random) selects the trajectory randomly.

| Experiment | Metric | DPCC-T | DPCC-C | DPCC-R | Ours |
|---|---|---|---|---|---|
| Hopper (Med-Expert) | safety | 0.01±0.03 | 0.02±0.02 | 0.02±0.02 | **0.00±0.00** |
| | admissib. | **0.00±0.00** | **0.00±0.00** | **0.00±0.00** | **0.00±0.00** |
| | dyn. consist. | **0.01±0.01** | **0.01±0.00** | **0.01±0.00** | **0.01±0.01** |
| | reward | 0.61±0.07 | 0.65±0.16 | 0.64±0.25 | **0.93±0.23** |
| | time | 4.24±1.64 | 4.56±1.72 | 6.21±2.61 | **0.06±0.00** |

early and aggressive control leads to sample deviation from the distribution learned by the generative model. In contrast, SAD-Flower activates control later using a lightweight QP-based formulation and preserves both safety and performance, highlighting the advantages of flexible, prescribed-time constraint enforcement.

# D  ENVIRONMENT DETAILS

We evaluate our method across a variety of trajectory planning domains, summarized in Table 8. The benchmark includes navigation (Maze2d-Umaze, Maze2d-Large), locomotion (Hopper, Walker2d), and robotic manipulation (Kuka Block-Stacking). Environments are simulated using MuJoCo (Todorov et al., 2012) or PyBullet (Coumans & Bai, 2016), and cover a range of state and action dimensions.

We use offline datasets to train the generative and control models for each benchmark. As shown in Table 9, the datasets for locomotion and navigation tasks are retrieved from the D4RL benchmark suite (Fu et al., 2020), with varying sizes (e.g., medium vs. medium-expert settings). For locomotion environments, datasets are collected using soft actor-critic (SAC) policies (Haarnoja et al., 2018), either partially trained or mixed with expert-level demonstrations. For the Kuka block-stacking task (Janner et al., 2022), data is generated via the PDDLStream planner (Garrett et al., 2020), which provides feasible robotic manipulation trajectories in structured stacking scenarios.

Table 8: Settings for each tasks.

| Environment | Simulator | Obs. Dim. | Action Dim. |
|---|---|---|---|
| Maze2d-Umaze-v1 | MuJoCo | 4 | 2 |
| Maze2d-Large-v1 | MuJoCo | 4 | 2 |
| Hopper | MuJoCo | 11 | 3 |
| Walker2d | MuJoCo | 17 | 6 |
| Kuka Block-Stacking | PyBullet | 39 | - |

Table 9: Dataset details for each benchmark environment, including the number of trajectories and the source or algorithm used to generate the data.

| Environment | # of Trajectories | Source / Generation Method |
|---|---|---|
| Maze2d-Umaze-v1 | $10^6$ | D4RL (Fu et al., 2020) |
| Maze2d-Large-v1 | $4 \times 10^6$ | D4RL (Fu et al., 2020) |
| Hopper-medium | $10^6$ | Partially trained SAC (Haarnoja et al., 2018) |
| Hopper-medium-expert | $2 \times 10^6$ | Mixture of expert and partial SAC |
| Walker2d-medium | $10^6$ | Partially trained SAC (Haarnoja et al., 2018) |
| Walker2d-medium-expert | $2 \times 10^6$ | Mixture of expert and partial SAC |
| Kuka Block-stacking | 10,000 | PDDLStream planner (Garrett et al., 2020) |

# E   CONSTRAINT SETTINGS IN EACH TASKS

We introduce task-specific state and action constraints to evaluate the ability of generative planners to handle safety and admissibility under diverse and challenging conditions. The test-time constraints are deliberately chosen to be more restrictive and often unseen during training, making constraint satisfaction a non-trivial requirement.

**Maze2d.**  The goal in Maze2d environments is to generate a feasible trajectory from a randomly sampled initial position to a randomly sampled goal location. The generative model is conditioned on both endpoints and produces full trajectories through the maze.

To evaluate constraint satisfaction, we introduce two novel, unseen obstacles at test time that do not completely block the path but add non-trivial planning constraints. The first is a superellipse-shaped obstacle defined as:

$$\left( \frac{x - x_0}{a} \right)^2 + \left( \frac{y - y_0}{b} \right)^2 \geq 1, \tag{10}$$

where $(x, y) \in \mathbb{R}^2$ is a trajectory state and $(x_0, y_0) \in \mathbb{R}^2$ is the center of the obstacle. The parameters $a > 0$ and $b > 0$ control the width and height of the obstacle.

The second is a higher-order polynomial barrier defined as:

$$\left( \frac{x - x_0}{a} \right)^4 + \left( \frac{y - y_0}{b} \right)^4 \geq 1. \tag{11}$$

This formulation results in sharper obstacle boundaries and makes naive post-processing or trajectory truncation ineffective due to the nonlinearity of the feasible region.

The action constraint is also tightened during test time by imposing a box constraint on the control inputs:

$$\boldsymbol{a} \in [-0.1, 0.1]^2. \tag{12}$$

Both `Maze2d-Umaze-v1` and `Maze2d-Large-v1` contain these two novel obstacles at test time to assess generalization to unseen constraints.

**Hopper and Walker2d.**  In the locomotion tasks, we impose test-time constraints that limit the robot's vertical motion to avoid collisions with overhead obstacles. Specifically, the height of the robot's torso must remain below a fixed roof height $z$, defined as:

$$s < z, \tag{13}$$

where $s$ is the vertical position of the torso. We evaluate this constraint under increasingly restrictive settings, such as $z = 1.6$, $1.4$, and $1.25$, to assess robustness to constraint tightening.

This state constraint is often in conflict with the task objective, which rewards forward jumping or walking. As a result, satisfying the constraint typically requires sacrificing task performance. Additionally, the control inputs are constrained by an action box constraint:

$$\boldsymbol{a} \in [-1, 1]^d, \tag{14}$$

where $d$ is the dimensionality of the action space (3 for Hopper and 6 for Walker2d).

**Kuka Block-Stacking.** For the Kuka block-stacking task, the generative model is conditioned on object locations and outputs joint trajectories for manipulation. To simulate partial system degradation or workspace reconfiguration, we apply a tighter state constraint on the robot's joint positions. Specifically, we scale the original feasible joint limits by a factor of 0.9:

$$\boldsymbol{q} \in 0.9 \cdot [\boldsymbol{q}_{\min}, \boldsymbol{q}_{\max}], \tag{15}$$

where $\boldsymbol{q}_{\min}$ and $\boldsymbol{q}_{\max}$ represent the original lower and upper bounds of each joint. No explicit action constraint is applied in this task.

These constraints, particularly when unseen during training, pose a significant challenge for planning and provide a rigorous benchmark for evaluating constraint satisfaction and generalization capabilities.

# F    TRAINING DETAIL

We provide implementation details of all baseline methods and our proposed approach, including training configurations, hyperparameters, and model architectures.

All baseline methods are trained using their official or publicly available codebases: Diffuser follows the implementation of Janner et al. (2022), Truncation, Classifier Guidance, and SafeDiffuser are implemented based on Xiao et al. (2025), Decision Diffuser follows Ajay et al. (2023), and Flow Matching uses the codebase from Feng et al. (2025). Our method is built on top of the same Flow Matching framework from Feng et al. (2025).

**Hyperparameters.** Table 10 shows the training hyperparameters for all baselines and our method across different environments. All methods use the same horizon and batch size per environment. Note that our method shares training settings with the flow matching baseline to ensure a fair comparison.

**Forward Dynamics Model Training.** To support constraint enforcement in SAD-Flower, we train a forward dynamics model for each environment using the same dataset used to train the generative models. This ensures a fair comparison without introducing additional supervision. The training details are summarized in Table 11.

For the Maze2d environments, a known analytical dynamics modelcan also be derived:

$$\begin{bmatrix} x(k+1) \\ y(k+1) \\ v_x(k+1) \\ v_y(k+1) \end{bmatrix} = \begin{bmatrix} 1 & 0 & dt & 0 \\ 0 & 1 & 0 & dt \\ 0 & 0 & 1 & 0 \\ 0 & 0 & 0 & 1 \end{bmatrix} \begin{bmatrix} x(k) \\ y(k) \\ v_x(k) \\ v_y(k) \end{bmatrix} + \begin{bmatrix} 0.5\alpha dt^2 & 0 \\ 0 & 0.5\alpha dt^2 \\ \alpha dt & 0 \\ 0 & \alpha dt \end{bmatrix} \begin{bmatrix} u_x(k) \\ u_y(k) \end{bmatrix} \tag{16}$$

where $[x_k, y_k, v_{x,k}, v_{y,k}]^T$ is the system state representing position and velocity, $[u_{x,k}, u_{y,k}]^T$ is the input force, $dt$ is the simulation time step, and $\alpha$ is the gear ratio divided by mass (due to primitive joint control).

**Model Architecture.** Diffusion-based models (Diffuser, Truncation, Classifier Guidance, SafeDiffuser) use the U-Net architecture with residual temporal convolutions, group normalization, and Mish nonlinearities, as described in (Janner et al., 2022). Flow Matching and SAD-Flower use the Transformer-based backbone proposed by (Feng et al., 2025), consisting of 8 layers with a hidden dimension of 256. The forward model in SAD-Flower is a feedforward neural network with 3 hidden layers of size 512, where the input dimension is the sum of the observation dimension and the action dimension, and the output is the observation dimension.

Table 10: Training hyperparameters for each method across environments.

| Maze2d-Umaze | Diffuser | Truncation | CG | S-Diffuser | FM | SAD-Flower |
|---|---|---|---|---|---|---|
| Batch Size | 32 | 32 | 32 | 32 | 32 | 32 |
| Learning Rate | $2e^{-4}$ | $2e^{-4}$ | $2e^{-4}$ | $2e^{-4}$ | $2e^{-4}$ | $2e^{-4}$ |
| Steps | $2e^6$ | $2e^6$ | $2e^6$ | $1e^6$ | $1e^6$ | $1e^6$ |
| **Maze2d-Large** | Diffuser | Truncation | CG | S-Diffuser | FM | SAD-Flower |
| Batch Size | 32 | 32 | 32 | 32 | 32 | 32 |
| Learning Rate | $2e^{-4}$ | $2e^{-4}$ | $2e^{-4}$ | $2e^{-4}$ | $2e^{-4}$ | $2e^{-4}$ |
| Steps | $2e^6$ | $2e^6$ | $2e^6$ | $1e^6$ | $1e^6$ | $1e^6$ |
| **Hopper** | Diffuser | Truncation | CG | S-Diffuser | FM | SAD-Flower |
| Batch Size | 32 | 32 | 32 | 32 | 32 | 32 |
| Learning Rate | $2e^{-4}$ | $2e^{-4}$ | $2e^{-4}$ | $2e^{-4}$ | $2e^{-4}$ | $2e^{-4}$ |
| Steps | $2e^6$ | $2e^6$ | $2e^6$ | $1e^6$ | $1e^6$ | $1e^6$ |
| **Walker2d** | Diffuser | Truncation | CG | S-Diffuser | FM | SAD-Flower |
| Batch Size | 32 | 32 | 32 | 32 | 32 | 32 |
| Learning Rate | $2e^{-4}$ | $2e^{-4}$ | $2e^{-4}$ | $2e^{-4}$ | $2e^{-4}$ | $2e^{-4}$ |
| Steps | $2e^6$ | $2e^6$ | $2e^6$ | $1e^6$ | $1e^6$ | $1e^6$ |
| **Kuka Block-Stacking** | Diffuser | Truncation | CG | S-Diffuser | FM | SAD-Flower |
| Batch Size | 32 | 32 | 32 | 32 | 32 | 32 |
| Learning Rate | $2e^{-5}$ | $2e^{-5}$ | $2e^{-5}$ | $2e^{-5}$ | $2e^{-4}$ | $2e^{-4}$ |
| Steps | $7e^5$ | $7e^5$ | $7e^5$ | $7e^5$ | $7e^5$ | $7e^5$ |

Table 11: Training settings for the forward dynamics models used in SAD-Flower.

| Environment | Batch Size | Learning Rate | Steps |
|---|---|---|---|
| Maze2d (Umaze/Large) | 256 | $1 \times 10^{-3}$ | $1 \times 10^6$ |
| Hopper | 256 | $1 \times 10^{-3}$ | $1 \times 10^6$ |
| Walker2d | 256 | $1 \times 10^{-3}$ | $1 \times 10^6$ |

# G COMPUTATIONAL RESOURCES

All experiments were conducted using four high-performance workstations with identical or near-identical configurations. Each workstation is equipped with an AMD EPYC series CPU, an NVIDIA Tesla P100 GPU, and 16 GB of GPU memory. The specific hardware details are summarized in Table 12.

Table 12: Hardware specifications of workstations used for training and evaluation.

| Workstation | CPU | GPU | GPU RAM |
|---|---|---|---|
| 1 | AMD EPYC 7542 | NVIDIA Tesla P100 | 16 GB |
| 2 | AMD EPYC 7542 | NVIDIA Tesla P100 | 16 GB |
| 3 | AMD EPYC 7742 | NVIDIA Tesla P100 | 16 GB |
| 4 | AMD EPYC 7542 | NVIDIA Tesla P100 | 16 GB |

# H THE USE OF LARGE LANGUAGE MODELS (LLMS)

Large language models (LLMs), e.g., ChatGPT, were employed to assist with editing, rephrasing, and LaTeX formatting during the paper writing process. No LLMs were involved in formulating research ideas, theoretical developments, or core experimental designs.

## I   COMPUTATION ANALYSIS

We report the detailed computational costs of SAD-Flower and all baseline methods across our benchmark tasks in Table 13. As expected, SAD-Flower incurs greater computational cost compared to unconstrained generative planners such as Diffuser and FM, due to the overhead introduced by enforcing multiple constraints during sampling. Nonetheless, in some domains—such as Hopper-Medium-Expert and Kuka Block-Stacking—SAD-Flower achieves lower runtime than SafeDiffuser, highlighting the efficiency gains enabled by our prescribed-time control formulation. Importantly, SAD-Flower is the only method that consistently satisfies all three constraint classes—state, action, and dynamic consistency—across all evaluated domains. No faster baseline provides this level of safety and reliability.

Table 13: Computation analysis of the proposed SAD-Flower and baselines across navigation, locomotion, and manipulation tasks.

| Experiment | Metric | Diffuser | Trunc | CG | FM | S-Diffuser | D-Diffuser | Ours |
|---|---|---|---|---|---|---|---|---|
| Maze2d (Large) | comp. time | 0.01±0.01 | 0.02±0.01 | 0.04±0.01 | 0.06±0.02 | 0.06±0.01 | 0.01±0.01 | 0.13±0.01 |
| Maze2d (Umaze) | comp. time | 0.01±0.01 | — | 0.04±0.01 | 0.02±0.01 | 0.06±0.02 | 0.01±0.01 | 0.08±0.02 |
| Hopper (Med-Expert) | comp. time | 0.05±0.01 | 0.06±0.02 | 0.06±0.01 | 0.01±0.01 | 0.11±0.02 | 0.02±0.01 | 0.06±0.01 |
| Hopper (Medium) | comp. time | 0.05±0.01 | 0.06±0.01 | 0.06±0.02 | 0.01±0.01 | 0.09±0.02 | 0.02±0.01 | 0.12±0.02 |
| Walker2D (Med-Expert) | comp. time | 0.06±0.02 | 0.06±0.01 | 0.05±0.01 | 0.01±0.01 | 0.09±0.02 | 0.02±0.01 | 0.09±0.01 |
| Walker2D (Medium) | comp. time | 0.04±0.01 | 0.13±0.03 | 0.21±0.02 | 0.01±0.01 | 0.10±0.02 | 0.02±0.01 | 0.09±0.01 |
| KUKA Block Stacking | comp. time | 0.70±0.01 | 0.84±0.01 | 0.86±0.01 | 0.47±0.06 | 0.78±0.01 | 0.04±0.01 | 0.74±0.08 |

## J   ADDITIONAL RELATED WORK

**Diffusion planning with reliability and constraint enforcement.**   Recent works have sought to improve the reliability of diffusion-based planners by modifying sampling strategies or adding post-hoc corrections, without addressing dynamic consistency or unseen constraints. Lee *et al.* propose *Refining Diffusion Planner for Reliable Behavior Synthesis* and introduce a restoration–gap metric predicted by a gap predictor to identify error-prone plans (Lee et al., 2023). The restoration gap acts as refining guidance, and an attribution-map regularizer prevents adversarial guidance, improving feasibility without altering the core diffusion model.

Feng *et al.* address stochastic failure modes by aggregating multiple trajectory samples. Their *Trajectory Aggregation Tree* (TAT) combines historical and current trajectories into a tree structure where each branch corresponds to a trajectory and nodes correspond to individual states (Feng et al., 2024). Unreliable states are marginalized and the most impactful nodes prioritized, yielding a training-free module deployable without modifying the original diffusion planner.

Wang *et al.* examine human-in-the-loop alignment. Their *Inference-Time Policy Steering* (ITPS) framework injects human interactions into the sampling process of a pretrained policy rather than fine-tuning on new data Wang et al. (2025). ITPS evaluates various forms of human interaction and sampling strategies to bias the denoising process toward user-desired subgoals while mitigating distribution shift.

LoMAP, introduced by Lee and Choi, targets off-manifold drift in guidance. The authors derive a lower bound on the guidance gap that quantifies manifold deviation and propose a training-free *Local Manifold Approximation and Projection* method Lee & Choi (2025). By projecting the guided sample onto a low-rank subspace approximated from offline data, LoMAP prevents infeasible trajectory generation and can be incorporated as a plug-in module for hierarchical diffusion planners.

Finally, Liang *et al.* extend diffusion planning to multi-robot settings. Their *Simultaneous Multi-Robot Motion Planning with Projected Diffusion Models* integrates constrained optimization into every denoising step, producing collision-free and kinematically feasible trajectories in dense multi-robot environments Liang et al. (2025). The authors introduce a comprehensive benchmark and report higher success rates compared with classical and learning-based planners.

These methods improve reliability by detecting artifacts, aggregating multiple candidates, steering policies with human input, projecting onto data manifolds, or enforcing per-step feasibility.

However, none simultaneously enforce safety, admissibility, and dynamic consistency, nor guarantee adherence to constraints absent from training data. By contrast, *SAD-Flower* formulates diffusion sampling as a controlled dynamical system. A quadratic-programming controller derived from control-barrier and control-Lyapunov conditions injects guidance at each denoising step, ensuring constraint satisfaction and dynamic consistency throughout the entire sampling horizon.

## K  DEPLOYMENT ON REAL-WORLD ROBOTIC PLATFORMS

Recent works have demonstrated the successful deployment of diffusion- and flow-based models on physical robotic systems (Chi et al., 2023; Sochopoulos et al., 2025), highlighting the feasibility of integrating such models into real-world robotic pipelines that involve perception, state estimation, and control. These results indicate that generative trajectory models can operate effectively outside of simulation.

SAD-Flower is designed as an open-loop planner that generates full trajectories prior to execution, without requiring real-time feedback control. This formulation avoids the need to meet strict control-frequency constraints, making it amenable to practical deployment. Although SAD-Flower solves a small quadratic program (QP) at each sampling step to enforce constraints, this overhead remains moderate and does not hinder offline trajectory generation. Moreover, recent advances in efficient generative planning—such as DiffuserLite (Dong et al., 2024) and Habi (Lu et al., 2025)—can be directly incorporated into our framework to further accelerate sampling, enabling tighter planning budgets without sacrificing constraint guarantees.

While our current implementation is focused on simulation, the modular structure of SAD-Flower makes it compatible with standard deployment pipelines. The forward model used for dynamics consistency can be learned directly from onboard sensor data, and constraint formulations (e.g., actuator limits, workspace boundaries) can be adapted to real-world specifications. As with all generative models, care must be taken to address sensor noise and execution uncertainty, but our control-theoretic formulation inherently provides robustness by enforcing constraints throughout the trajectory.

In summary, SAD-Flower introduces only modest additional computation to existing generative planners while providing formal guarantees for safety, admissibility, and dynamic consistency. Given the demonstrated deployability of similar frameworks and the compatibility of our method with efficient sampling and standard control pipelines, we believe SAD-Flower can be incorporated into real-world robotic systems with limited adaptation. Investigating such deployment remains an important direction for future work.

## L  COMPARISON WITH CONSTRAINT-AWARE BASELINES

We further compare SAD-Flower against two additional constraint-aware baselines: a reject-sampling method, which offers a simple heuristic for constraint handling, and CoBL-Diffusion (Mizuta & Leung, 2024), which leverages control-theoretic rewards to encourage constraint satisfaction.

The reject-sampling approach first trains a flow matching model on the original dataset, which includes trajectories that violate test-time constraints, since those constraints are unseen during training. At inference, it samples a batch of trajectory candidates and discards those violating constraints, selecting from the remaining set the trajectory with the least violation. As shown in Table 14, this approach consistently fails to satisfy constraints across all tasks, including Maze2d, Hopper, Walker2d, and Kuka Block-Stacking. Both safety and admissibility violations remain non-zero. This confirms that post-hoc filtering alone is insufficient when the generative model has no knowledge of constraint structure—especially under novel test-time constraints. Even though flow matching can model multimodal behavior patterns, it cannot reliably sample within unseen constraint sets unless such constraints are explicitly incorporated during inference.

CoBL-Diffusion injects physical constraints into a diffusion model by shaping its denoising process with auxiliary rewards derived from Control Barrier Functions (CBFs) and Control Lyapunov Functions (CLFs). This allows it to softly bias generation toward constraint-adherent behaviors, but without any formal guarantees. We evaluate CoBL-Diffusion in the LargeMaze environment, using

the same horizon and constraints as in other baselines. As shown in Table 15, the method fails to fully enforce safety and admissibility constraints. This is expected given several key limitations: (1) it predicts only actions, whereas SAD-Flower jointly generates states and actions, which allows for more precise control of trajectory behavior; (2) its CBFs target only state constraints, while ours apply to both state and action spaces; (3) it uses CLFs for general stability, not for ensuring formal dynamic consistency as we do; and (4) it lacks prescribed-time scheduling, making it difficult to guarantee constraint satisfaction at the final step. These design choices limit CoBL-Diffusion's ability to robustly handle complex, temporally structured constraints. In contrast, SAD-Flower provides a framework to reliably enforce constraints even under novel test-time scenarios.

Table 14: Performance of SAD-Flower and reject-sample method across all tasks.

| Experiment | Metric | Reject-sample | Ours |
|---|---|---|---|
| Maze2d (Large) | safety | $0.16 \pm 0.29$ | $\mathbf{0.00 \pm 0.00}$ |
| | admissib. | $0.90 \pm 0.01$ | $\mathbf{0.00 \pm 0.00}$ |
| | dyn. consist. | $0.02 \pm 0.01$ | $\mathbf{0.01 \pm 0.01}$ |
| | reward | $\mathbf{1.52 \pm 0.28}$ | $1.42 \pm 0.52$ |
| Maze2d (Umaze) | safety | $0.04 \pm 0.09$ | $\mathbf{0.00 \pm 0.00}$ |
| | admissib. | $0.91 \pm 0.01$ | $\mathbf{0.00 \pm 0.00}$ |
| | dyn. consist. | $\mathbf{0.01 \pm 0.01}$ | $\mathbf{0.01 \pm 0.01}$ |
| | reward | $\mathbf{2.91 \pm 0.65}$ | $2.66 \pm 0.88$ |
| Hopper (Med-Expert) | safety | $0.14 \pm 0.18$ | $\mathbf{0.00 \pm 0.00}$ |
| | admissib. | $0.05 \pm 0.02$ | $\mathbf{0.00 \pm 0.00}$ |
| | dyn. consist. | $0.02 \pm 0.01$ | $\mathbf{0.01 \pm 0.01}$ |
| | reward | $0.49 \pm 0.33$ | $\mathbf{0.93 \pm 0.23}$ |
| Hopper (Medium) | safety | $0.03 \pm 0.04$ | $\mathbf{0.00 \pm 0.00}$ |
| | admissib. | $0.06 \pm 0.03$ | $\mathbf{0.00 \pm 0.00}$ |
| | dyn. consist. | $0.02 \pm 0.02$ | $\mathbf{0.01 \pm 0.01}$ |
| | reward | $0.24 \pm 0.14$ | $\mathbf{0.34 \pm 0.03}$ |
| Walker2d (Med-Expert) | safety | $0.08 \pm 0.10$ | $\mathbf{0.00 \pm 0.00}$ |
| | admissib. | $0.08 \pm 0.08$ | $\mathbf{0.00 \pm 0.00}$ |
| | dyn. consist. | $\mathbf{0.04 \pm 0.06}$ | $\mathbf{0.04 \pm 0.04}$ |
| | reward | $\mathbf{0.97 \pm 0.22}$ | $0.89 \pm 0.32$ |
| Walker2d (Medium) | safety | $0.18 \pm 0.15$ | $\mathbf{0.00 \pm 0.00}$ |
| | admissib. | $0.16 \pm 0.10$ | $\mathbf{0.00 \pm 0.00}$ |
| | dyn. consist. | $\mathbf{0.07 \pm 0.17}$ | $\mathbf{0.07 \pm 0.15}$ |
| | reward | $\mathbf{0.68 \pm 0.19}$ | $0.42 \pm 0.23$ |
| KUKA Block Stacking | safety | $0.01 \pm 0.01$ | $\mathbf{0.00 \pm 0.00}$ |
| | reward | $0.44 \pm 0.67$ | $\mathbf{0.45 \pm 0.21}$ |

Table 15: Performance of SAD-Flower and CoBL-Diffusion in the navigation task.

| Experiment | Metric | CoBL-Diffusion | Ours |
|---|---|---|---|
| Maze2d (Large) | safety | $0.01 \pm 0.04$ | $\mathbf{0.00 \pm 0.00}$ |
| | admissib. | $0.23 \pm 0.33$ | $\mathbf{0.00 \pm 0.00}$ |
| | dyn. consist. | $\mathbf{0.00 \pm 0.00}$ | $0.01 \pm 0.01$ |
| | reward | $0.15 \pm 0.18$ | $\mathbf{1.42 \pm 0.52}$ |

