# OpenReview forum: "SAD-Flower: Flow Matching for Safe, Admissible, and Dynamically Consistent Planning"
_ICLR.cc/2026/Conference — Submitted to ICLR 2026_

### Official Review · Reviewer_KkJf · 2025-10-18

**Soundness:** 3
**Presentation:** 3
**Contribution:** 3
**Rating:** 6
**Confidence:** 4

**Summary:**

The author introduces SAD-Flower, a framework for generating safe, admissible, and dynamically consistent trajectories using flow matching (FM). The core problem addressed is that standard generative planners, including those based on FM, lack formal guarantees for satisfying state, action, and dynamics constraints. SAD-Flower reframe the trajectory generation process as a controllable dynamical system by augmenting the learned vector field of the FM with a virtual control input. The authors provide theoretical guarantees for constraint satisfaction and demonstrate empirically across several benchmarks that SAD-Flower significantly outperforms baselines in constraint adherence while maintaining competitive task performance.

**Strengths:**

1. SAD-Flower is the first, to my knowledge, to provide a unified framework for safety, admissibility, and dynamic consistency with formal guarantees within a flow matching context.
2. The paper is well-structured and easy to follow.
3. The proposed method is theoretically well-grounded.

**Weaknesses:**

1. A potential weakness is that the experiments are conducted in environments with relatively low-dimensional state and action spaces. Since the CLF formulation relies on an accurate dynamics model, it is unclear how well the proposed approach would generalize to more challenging, high-dimensional environments, such as the OGBench Humanoid benchmark, where learning an accurate dynamics model is notoriously difficult.
2. There is limited analysis of how computational cost scales with problem complexity like horizon length, the number of constraint and state/action dimensionality.
3. Many recent works have explored diffusion sampling that satisfies constraints and ensures dynamic consistency. However, the current paper overlooks these relevant works. I encourage the authors to include a discussion of the following references like:

[1] Refining Diffusion Planner for Reliable Behavior Synthesis by Automatic Detection of Infeasible Plans, 2023

[2] Resisting stochastic risks in diffusion planners with the trajectory aggregation tree, 2024

[3] Inference-Time Policy Steering through Human Interactions, 2025

[4] Local Manifold Approximation and Projection for Manifold-Aware Diffusion Planning, 2025

[5] Simultaneous Multi-Robot Motion Planning with Projected Diffusion Models, 2025

**Questions:**

See weaknesses

---

> ### Author Response · Authors · 2025-11-24
> **Response to review comments (1/2)**
>
> We are grateful to Reviewer KkJf for the positive and insightful review. We appreciate your recognition that SAD-Flower introduces, to the best of your knowledge, the first unified framework that jointly handles safety, admissibility, and dynamic consistency with formal guarantees in a flow-matching context. We are also glad that you found the presentation clear and the theoretical formulation well-grounded. Your comments on scaling to higher-dimensional domains and situating the work relative to recent diffusion-based planners are particularly helpful, and we expand on these points in the discussion below.
>
> > ***W1***: *The experiments focus on relatively low-dimensional environments, raising concerns about whether the CLF-based method would generalize to high-dimensional domains where accurate dynamics learning is difficult.*
>
> **Response:**
>
> Thank you for highlighting the question of scalability to high-dimensional systems. We agree that this is an important consideration.
>
> First, our experiments already include Walker2D (17-dimensional state, 6-dimensional action), which demonstrates that SAD-Flower can operate beyond simple low-dimensional settings.
>
> Second, SAD-Flower does not inherently rely on learned dynamics models. When an accurate dynamics model is available—for example in many locomotion and robot-arm settings where high-fidelity simulation frameworks exist [1–3]—our dynamic-consistency guarantee is fully rigorous. When a learned model must be used, the guarantee becomes approximate, but its behavior is well understood: model error translates directly into consistency error, as formalized in the model-based RL literature (e.g., [4]). In practice, we observe only very small consistency violations, confirming that the learned models used in our experiments are accurate enough for reliable planning.
>
>
> Third, to directly address the reviewer’s concern about scalability, we **added a new experiment on dexterous grasping** using the **D4RL Adroit Relocate benchmark (39D state, 30D action)**.  We define the state constraint by the arm's position for obstacle avoidance, and the action constraint by the actuator limits. This is a substantially more challenging, contact-rich, high-dimensional manipulation task. The full experimental setup and results have been **added to `Section 6.4`** in the revised paper. As shown in Table A, FM violates safety and admissibility constraints, while **SAD-Flower satisfies both perfectly**, with only minor numerical deviations in dynamic consistency.
>
> Table A:  Performance of SAD-Flower and FM in the dexterous grasping scenario
>
> | Method | Safety($\downarrow$) | Admissibility($\downarrow$) | Dynamic consistency($\downarrow$) | score($\uparrow$) |
> | -------- | -------- | -------- | -------- | -------- |
> | FM     | $0.15 \pm 0.21$     | $0.62 \pm 0.19$     | $0.07 \pm 0.04$ | $1.07 \pm 0.08$     |
> | SAD-Flower     | $\boldsymbol{0.00 \pm 0.00}$     | $\boldsymbol{0.00 \pm 0.00}$     | $\boldsymbol{0.06 \pm 0.09}$ | $1.05 \pm 0.05$     |
>
> *[1] C. Gaz, M. Cognetti, A. Oliva, P. R. Giordano, and A. De Luca, “Dynamic identification of the franka emika panda robot with retrieval of feasible parameters using penalty-based optimization,” IEEE Robotics and Automation Letters, 4(4), 4147-4154, 2019.*
>
> *[2] T. A. Howell, S. L. Cleac'h, J. Brüdigam, J. Z. Kolter, M. Schwager, and Z. Manchester, “Dojo: A differentiable physics engine for robotics,” arXiv preprint arXiv:2203.00806, 2022.*
>
> *[3] B. Acosta, W. Yang, and M. Posa, "Validating robotics simulators on real-world impacts," IEEE Robotics and Automation Letters 7.3: 6471-6478, 2022.*
>
> *[4] C. Sebastian, F. Berkenkamp, and A. Krause. "Efficient model-based reinforcement learning through optimistic policy search and planning." Advances in Neural Information Processing Systems, 14156-14170, 2020.*

---

> ### Author Response · Authors · 2025-11-24
> **Response to review comments (2/2)**
>
> > ***W2***: *There is little analysis of how computational cost scales with horizon length, number of constraints, or state and action dimensionality.*
>
> **Response:**
>
> Thank you for the question. To address this, we provide in Table B the planning horizons, state/action dimensionalities, and average computation times across all evaluated domains. These results have also been included in `Appendix I` of the revised paper, with additional simulation details in `Sections 6.1–6.2` and `Appendices D–E`. The numbers reveal only a mild growth in computation time as the problem complexity increases. For example, comparing UMaze (H = 128) and LargeMaze (H = 384) shows that a 3× increase in horizon leads to only a ~40 ms increase in QP solve time. Likewise, moving from low-dimensional navigation (4D state, 2D action) to high-dimensional manipulation (39D state, 7D action) increases the cost to ~0.74 s, which remains manageable given the significantly higher dimensionality.
>
> Table B: Computational cost and settings for SAD-Flower
>
> | Metric | UMaze | LargeMaze | Hopper(Med-Expert) | Hopper(Medium) | Walker2d(Med-Expert) | Walker2d(Medium) | KUKA Block Stacking |
> |:------:|:-----:|:----------:|:-------------------:|:----------------:|:----------------------:|:------------------:|:---------------------:|
> | Horizon | $128$ | $384$ | $50$ | $50$ | $50$ | $50$ | $128$ |
> | State Dim | $4$ | $4$ | $11$ | $11$ | $17$ | $17$ | $39$ |
> | Act Dim | $2$ | $2$ | $3$ | $3$ | $6$ | $6$ | $7$ |
> | Time(s) | $0.09 \pm 0.02$ | $0.13 \pm 0.01$ | $0.06 \pm 0.01$ | $0.12 \pm 0.01$ | $0.09 \pm 0.01$ | $0.09 \pm 0.01$ | $0.74 \pm 0.08$ |
>
>
> > ***W3***: *The paper overlooks several recent works on diffusion-based planning that address constraint satisfaction and dynamic consistency*
>
>
> **Response:**
>
> We thank the reviewer for pointing us to these recent works on diffusion-based planning. **We have discussed them explicitly in `Appendix J` to situate SAD-Flower more clearly within the literature**. Each cited approach improves the reliability of generative planners, but **none of them aim to guarantee constraint satisfaction under constraints that were absent from the training data**, which is the central challenge addressed in our work.
>
>
> Each of the cited papers addresses reliability or feasibility in generative planning but does so through a different mechanism. Refining Diffusion Planner [1] for Reliable Behavior Synthesis introduces a restoration gap predictor to detect and correct unreliable plans without modifying the underlying model. Trajectory Aggregation Tree [2] aggregates and filters multiple samples to reduce stochastic failure modes. Inference Time Policy Steering [3] focuses on aligning a pre-trained policy with human inputs by combining alignment gradients with the denoising process. LoMAP [4] prevents off manifold drift by retrieving nearby dataset trajectories, constructing a local low rank approximation and projecting samples onto that subspace. Simultaneous Multi Robot Motion Planning with Projected Diffusion Models [5] incorporates a projection operator that solves a constrained optimization at every denoising step to ensure collision-free feasibility in multi-robot settings.
>
> These methods improve reliability through artifact detection, multi-sample filtering, human-in-the-loop steering, manifold projection, or stepwise projection-based feasibility, but
> • none jointly enforce state safety, action admissibility, and dynamic consistency,
> • none ensure correctness when constraints do not appear in the training distribution.
>
> By contrast, **SAD-Flower is specifically designed to enforce novel constraints at test time**, without retraining the generative model. Our formulation treats sampling as a controlled dynamical system and adds a QP-derived control input that enforces CBF and CLF conditions. This yields guaranteed adherence to state, action, and dynamic-consistency constraints—even under previously unseen conditions—something not addressed by existing diffusion-based planning approaches.
>
> *[1] K. Lee, Seongun Kim, and Jaesik Choi. "Refining diffusion planner for reliable behavior synthesis by automatic detection of infeasible plans." Advances in Neural Information Processing Systems, 36:24223–24246, 2023.*
>
> *[2] L. Feng, Pengjie Gu, Bo An, and Gang Pan. "Resisting stochastic risks in diffusion planners with the trajectory aggregation tree." arXiv preprint arXiv:2405.17879, 2024.*
>
> *[3] Y. Wang, L. Wang, Y. Du, B. Sundaralingam, X. Yang, Y. Chao, C. D’Arpino, D. Fox, and J. Shah. "Inference-time policy steering through human interactions." In 2025 ICRA, pp. 15626–15633. IEEE, 2025*
>
> *[4] K. Lee and J. Choi. "Local manifold approximation and projection for manifold-aware diffusion planning." arXiv preprint arXiv:2506.00867, 2025.*
>
> *[5] J. Liang, J. K Christopher, S. Koenig, and Ferdinando Fioretto. "Simultaneous multi-robot motion planning with projected diffusion models." arXiv preprint arXiv:2502.03607, 2025.*

---

### Official Review · Reviewer_ri9Z · 2025-10-30

**Soundness:** 2
**Presentation:** 3
**Contribution:** 3
**Rating:** 4
**Confidence:** 4

**Summary:**

This paper proposes SAD-Flower, a novel control-augmented flow matching framework to address the limitations of existing flow matching (FM) based planners in providing formal guarantees for safety, admissibility, and dynamic consistency of generated trajectories. By introducing a virtual control input and leveraging nonlinear control theory, specifically Control Barrier Functions (CBFs) and Control Lyapunov Functions (CLFs), SAD-Flower ensures constraint satisfaction and dynamic consistency without requiring retraining for unseen constraints. The framework uses a quadratic program (QP) to determine the minimum-norm control input. Extensive experiments on navigation, locomotion, and manipulation tasks demonstrate SAD-Flower's superior performance in adhering to constraints compared to various generative-model-based baselines.

**Strengths:**

1. The paper provides a rigorous theoretical foundation by integrating nonlinear control theory (CBFs and CLFs) to offer formal guarantees for safety, admissibility, and dynamic consistency. This is a significant improvement over existing FM-based methods that often lack such assurances.
2.  SAD-Flower's ability to enforce new or tighter constraints without requiring model retraining is a major practical advantage, enhancing its real-world applicability and robustness.
3. The comprehensive experimental evaluation across diverse and challenging environments (Maze2d, Hopper, Walker2d, Kuka Block-Stacking) consistently demonstrates SAD-Flower's superior performance in satisfying constraints and achieving higher rewards compared to several baselines.

**Weaknesses:**

1. The introduction of a virtual control input and the need to solve a QP at each step for the minimum-norm control input may introduce significant computational overhead, potentially limiting real-time application in very high-dimensional or time-critical systems. While the paper mentions "lightweight QP-based formulation," specific runtime comparisons or analysis of the QP's complexity for varying state/action spaces could strengthen this.
2. Defining appropriate CBFs and CLFs, especially for complex, high-dimensional real-world tasks, can be challenging and might require expert knowledge, potentially limiting the ease of adoption for new problem setups.
3. While experiments cover various tasks, the scalability of SAD-Flower to environments with extremely complex, non-smooth, or highly dynamic constraints, or very long horizons, could be further explored.

**Questions:**

1. Could the authors provide a more detailed analysis or empirical comparison of the computational cost (e.g., inference time per step) of SAD-Flower compared to baselines, specifically highlighting the overhead introduced by the QP solver for different problem complexities?
2. Are there any guidelines or automated methods proposed for the systematic design of CBFs and CLFs for new, complex tasks, or does it heavily rely on manual finetuning?
3. How sensitive is SAD-Flower's performance to the choice of hyperparameters, especially those related to the CBF/CLF formulation and the QP solver settings?
4. For long-horizon tasks or those with sparse rewards, trajectory optimization can be challenging. How does the "flow matching" aspect of SAD-Flower (which often relies on expert demonstrations or a pre-trained policy) perform in such scenarios? Does the control augmentation inherently improve performance in these difficult settings, or are there specific limitations?
5. How does SAD-Flower conceptually and practically compare to other hybrid control architectures that combine learning-based methods with formal verification or safety filters (e.g., using model predictive control (MPC) with safety guarantees)? What are the distinct advantages and disadvantages of SAD-Flower in relation to these alternative approaches?
6. Can the framework be extended to multi-agent planning scenarios where cooperative or adversarial interactions are present, and how would the constraint satisfaction and dynamic consistency guarantees be maintained in such complex settings?
7. Have there been any preliminary tests or considerations for deploying SAD-Flower on real-world robotic platforms, and what challenges might arise regarding sensor noise, actuation limits, and real-time execution?

---

> ### Author Response · Authors · 2025-11-24
> **Response to review comments (1/5)**
>
> We thank Reviewer ri9Z for the encouraging and detailed feedback. We are pleased that the theoretical integration of CBFs and CLFs within flow-matching planners stood out as a meaningful contribution, and that you viewed SAD-Flower’s ability to enforce new constraints without retraining as a strong practical advantage. We also appreciate your recognition of the breadth and consistency of our experimental results across navigation, locomotion, and manipulation tasks. The questions you pose regarding computational overhead, control-design methodology, and extensions to more complex settings give us clear opportunities to deepen the discussion, which we address in the following sections.
>
> > ***W2***: *Designing suitable CBFs and CLFs for complex, high-dimensional real-world tasks can be difficult and may require substantial expert knowledge, which could limit ease of adoption.*
>
> **Response:**
>
> Thank you for raising this concern. While designing CBFs and CLFs for complex, high-dimensional robotic systems can indeed be challenging in general, this difficulty is substantially reduced in SAD-Flower. Crucially, we do not define the CLF and CBF on the robot’s physical dynamics. Instead, they are defined on the flow-matching dynamics augmented with a virtual control input, which reduces to a simple integrator. This dramatically simplifies the construction of valid CBFs and CLFs. We show **(Theorem B.1)** that the **signed distance function already suffices to ensure feasibility for the CBF constraint**s alone under weak assumptions. Similarly, we prove **(Theorem B.2)** that **a quadratic CLF yields a constraint that is feasible under weak conditions** on the dynamics of the robot. Thus, feasibility of the individual constraints is not an issue. While we cannot exclude that the combination of CBF and CLF constraints causes the infeasibility of the QP, this requires the perfect alignment of two constraints, but pointing in opposite directions. Given the high-dimensional space because of the trajectories, this seems rather unlikely. This intuition is further supported by our numerical evaluation, where we did not observe any feasibility issues and a fallback strategy was not necessary. Note that **the relaxation using slack variables is already discussed briefly after Theorem 5.1** in the paper.
>
>
> > ***W3***: *Although the experiments span several tasks, the scalability of SAD-Flower to environments with extremely complex, non-smooth, highly dynamic constraints or very long horizons remains underexplored.*
>
> **Response:**
>
> We appreciate the reviewer’s insightful comment regarding the scope of our evaluation. We fully agree that demonstrating the applicability of our method in more complex tasks is valuable.
>
> To demonstrate the applicability of our method to more complex manipulation scenarios that involve challenging grasps and contact-rich dynamics, we **additionally evaluate on the D4RL Adroit Relocate dexterous grasping task (39-dimensional state, 30-dimensional action).**  We define the state constraint by the arm's position for obstacle avoidance, and the action constraint by the actuator limits  This high-dimensional, contact-rich manipulation scenario represents a substantially more challenging benchmark than the tasks previously considered. The full experimental setup and results have been **added to `Section 6.4` in the revised paper.**
>
> The results (see table A) highlight the core motivation of SAD-Flower: while standard flow matching (FM) achieves reasonable task performance, it exhibits notable violations of safety and admissibility constraints due to its unconstrained sampling. In contrast, **SAD-Flower reliably enforces both state and action constraints**, even in this challenging setting. Although dynamical consistency is not perfectly satisfied, the residual violations are small and primarily attributable to discretization and numerical integration.
>
> This experiment demonstrates that our method scales effectively to complex grasping domains and provides evidence that the proposed control-augmented mechanism meaningfully improves the reliability and safety of generative planners in high-dimensional robotic applications.
>
> Table A:  Performance of SAD-Flower and FM in the dexterous grasping scenario
> | Method | Safety($\downarrow$) | Admissibility($\downarrow$) | Dynamic consistency($\downarrow$) | score($\uparrow$) |
> | -------- | -------- | -------- | -------- | -------- |
> | FM     | $0.15 \pm 0.21$     | $0.62 \pm 0.19$     | $0.07 \pm 0.04$ | $1.07 \pm 0.08$     |
> | SAD-Flower     | $\boldsymbol{0.00 \pm 0.00}$     | $\boldsymbol{0.00 \pm 0.00}$     | $\boldsymbol{0.06 \pm 0.09}$ | $1.05 \pm 0.05$     |

---

> ### Author Response · Authors · 2025-11-24
> **Response to review comments (2/5)**
>
> > ***Q2***: *It is unclear whether there are systematic guidelines or automated methods for constructing CBFs and CLFs in new, complex tasks, or whether the approach relies heavily on manual tuning.*
>
> **Response:**
>
> Thank you for the question. As discussed earlier, SAD-Flower is specifically designed to avoid the difficulty of hand-engineering CLFs and CBFs for complex robotic dynamics. Because the CLF and CBF are defined on the flow-matching dynamics augmented with a virtual control input, the underlying system reduces to a simple integrator. This makes the construction of valid functions both systematic and largely task-independent.
>
> In practice, we use only two generic components:
>
> 1. a distance-based CBF, such as the signed distance to obstacles or actuator limits, and
> 2. a quadratic CLF that penalizes dynamic inconsistency.
>
> Thus, unlike traditional CBF/CLF design for complex nonlinear systems, SAD-Flower does not require expert manual construction or domain-specific tuning.
>
>
> > ***Q3***: *How sensitive is SAD-Flower's performance to the choice of hyperparameters, especially those related to the CBF/CLF formulation and the QP solver settings?*
>
> **Response:**
>
> We appreciate the reviewer’s question regarding hyperparameter sensitivity. One of our design goals for SAD-Flower is to avoid delicate tuning, and in practice the method has very few hyperparameters. The only term that influences the CBF/CLF formulation is the prescribed-time constant $c$, which controls how quickly the constraints tighten as $t\to 1$.
>
> To evaluate sensitivity, we conducted an additional ablation study on the LargeMaze task—now included in **`Section 6.4` of the revised paper**—by varying $c$ over a wide range (from $1.0$ down to $0.2$). As summarized in Table B, **SAD-Flower maintains perfect safety and admissibility** for all tested values. Dynamic-consistency deviations remain extremely small and stable, and the overall task score changes only modestly across these values. The QP setting for all these results follows the default setting in the qpth package [1].
>
> Table B: Sensitivity of SAD-Flower to the prescribed-time hyperparameter $c$
>
> | hyperparameters | Safety($\downarrow$) | Admissibility($\downarrow$) | Dynamic consistency($\downarrow$) | score($\uparrow$) |
> |:---------------:|:---------------------:|:----------------------------:|:----------------------------------:|:------------------:|
> | $1.0$ | $0.00 \pm 0.00$ | $0.00 \pm 0.00$ | $0.01 \pm 0.01$ | $1.38 \pm 0.69$ |
> | $0.9$ | $0.00 \pm 0.00$ | $0.00 \pm 0.00$ | $0.01 \pm 0.01$ | $1.37 \pm 0.58$ |
> | $0.8$ | $0.00 \pm 0.00$ | $0.00 \pm 0.00$ | $0.01 \pm 0.01$ | $1.54 \pm 0.59$ |
> | $0.7$ | $0.00 \pm 0.00$ | $0.00 \pm 0.00$ | $0.01 \pm 0.01$ | $1.61 \pm 0.23$ |
> | $0.6$ | $0.00 \pm 0.00$ | $0.00 \pm 0.00$ | $0.01 \pm 0.01$ | $1.46 \pm 0.52$ |
> | $0.5$ | $0.00 \pm 0.00$ | $0.00 \pm 0.00$ | $0.01 \pm 0.10$ | $1.56 \pm 0.57$ |
> | $0.4$ | $0.00 \pm 0.00$ | $0.00 \pm 0.00$ | $0.01 \pm 0.01$ | $1.47 \pm 0.53$ |
> | $0.3$ | $0.00 \pm 0.00$ | $0.00 \pm 0.00$ | $0.02 \pm 0.01$ | $1.57 \pm 0.24$ |
> | $0.2$ | $0.00 \pm 0.00$ | $0.00 \pm 0.00$ | $0.02 \pm 0.01$ | $1.29 \pm 0.73$ |
>
>
> [1] https://github.com/locuslab/qpth

---

> ### Author Response · Authors · 2025-11-24
> **Response to review comments (3/5)**
>
> > ***W1***: *Solving a QP with a virtual control input at every step may introduce non-trivial computational overhead, and clearer runtime or complexity analysis would help justify the claimed “lightweight” formulation in higher-dimensional or time-critical systems.*
>
> > ***Q1***: *A more detailed breakdown of computational cost, such as per-step inference time compared to baselines and the overhead specifically attributable to the QP solver across different task complexities, is needed.*
>
>
> **Response:**
>
> We sincerely thank the reviewer for raising this important point about computational overhead and for encouraging a clearer breakdown of runtime and complexity. We fully agree that understanding the cost of the QP component is crucial for assessing the practicality of SAD-Flower across domains of varying difficulty.
>
> First, although the QP is in principle solved at every integration step, **our prescribed-time scheduling significantly reduces the effective number of solves.** As $t \to 1$, the CBF and CLF constraints naturally become inactive and the optimal control collapses to $\boldsymbol{u}=0$. This means that many late-stage denoising steps require no optimization at all, reducing overhead without compromising constraint satisfaction.
>
> Second, the problems we consider **involve nonlinear dynamics with coupled state–action constraints**. Standard approaches for enforcing such constraints—such as MPC [1]—typically require repeatedly solving large nonlinear programs. Similarly, recent diffusion-based controllers such as DPCC [2] rely on nonlinear trajectory optimization to project state–action sequences onto the safe and admissible set. **These nonlinear programs are substantially more expensive than our QP-based formulation**, which is discussed in the “QP vs. Non-convex Optimization” paragraph of `Section 6.3`. In contrast, SAD-Flower requires only a lightweight QP. This QP framework is more computationally efficient while still providing guarantees on safety, admissibility, and dynamic consistency, as shown in the ablation study in Table 2 of the paper.
>
> To address the reviewer’s request, we provide detailed computational costs for all baselines and SAD-Flower across our benchmarks in Appendix I of the revised paper. We observe that SAD-Flower is indeed slower than purely unconstrained methods (e.g., Diffuser, FM), which is expected because it enforces multiple constraints. Nevertheless, **SAD-Flower sometimes can achieve lower computation times than SafeDiffuser**, e.g., Hopper-expert and KUKA, due to our prescribed-time formulation. More importantly, **SAD-Flower is the only method that satisfies all three forms of constraints—state, action, and dynamic consistency—across all domains**, a property no faster method provides. The part of runtime results (Due to the word limit) is included in Table B below, and the full set of results is included in `Appendix I`.
>
> Table B: Runtime and performance of SAD-Flower and baselines.
>
> | Experiment | Metric | Diffuser | Trunc | CG | FM | S-Diffuser | D-Diffuser | Ours |
> |:---:|:---:|:---:|:---:|:---:|:---:|:---:|:---:|:---:|
> | **Hopper (Med-Expert)** | **safety** | $0.01 \pm 0.02$ | $0.05 \pm 0.04$ | $0.07 \pm 0.03$ | $0.11 \pm 0.08$ | $0.05 \pm 0.04$ | $0.10 \pm 0.02$ | $\boldsymbol{0.00 \pm 0.00}$ |
> |  | **admissib.** | $0.21 \pm 0.05$ | $0.18 \pm 0.04$ | $0.26 \pm 0.07$ | $0.17 \pm 0.05$ | $0.18 \pm 0.04$ | $\boldsymbol{0.00 \pm 0.00}$ | $\boldsymbol{0.00 \pm 0.00}$ |
> |  | **dyn. consist.** | $0.38 \pm 0.03$ | $0.41 \pm 0.04$ | $0.79 \pm 0.10$ | $0.23 \pm 0.02$ | $0.36 \pm 0.06$ | $0.16 \pm 0.01$ | $\boldsymbol{0.01 \pm 0.01}$ |
> |  | **reward** | $1.06 \pm 0.18$ | $0.50 \pm 0.12$ | $0.73 \pm 0.02$ | $1.02 \pm 0.20$ | $0.53 \pm 0.19$ | $\boldsymbol{1.12 \pm 0.01}$ | $0.93 \pm 0.23$ |
> |  | **comp. Time** | $0.05 \pm 0.01$ | $0.06 \pm 0.02$ | $0.06 \pm 0.01$ | $0.01 \pm 0.01$ | $0.11 \pm 0.02$ | $0.02 \pm 0.01$ | $0.06 \pm 0.01$ |
> | **KUKA Block Stacking** | **safety** | $0.23 \pm 0.09$ | $\boldsymbol{0.00 \pm 0.00}$ | $0.22 \pm 0.09$ | $0.02 \pm 0.04$ | $\boldsymbol{0.00 \pm 0.00}$ | $0.14 \pm 0.13$ | $\boldsymbol{0.00 \pm 0.00}$ |
> |  | **reward** | $0.46 \pm 0.23$ | $0.45 \pm 0.21$ | $0.45 \pm 0.23$ | $0.44 \pm 0.20$ | $0.49 \pm 0.23$ | $\boldsymbol{0.55 \pm 0.26}$ | $0.45 \pm 0.21$ |
> |  | **comp. Time** | $0.70 \pm 0.01$ | $0.84 \pm 0.01$ | $0.86 \pm 0.01$ | $0.47 \pm 0.06$ | $0.78 \pm 0.01$ | $0.04 \pm 0.01$ | $0.74 \pm 0.08$ |
>
>
> *[1] J. B Rawlings, D. Q Mayne, M. Diehl, et al. Model predictive control: theory,
> computation, and design, volume 2. Nob Hill Publishing Madison, WI, 2017.*
>
> *[2] R. Römer, A. V Rohr, and A. P Schoellig. Diffusion predictive control with constraints. Learning for Dynamics and Control, 2025.*

---

> ### Author Response · Authors · 2025-11-24
> **Response to review comments (4/5)**
>
> > ***Q4***: *For long-horizon or sparse-reward tasks, it is not fully explained how the flow-matching component performs, particularly when trained from expert data, and whether the control augmentation improves performance or introduces limitations.*
>
> **Response:**
>
> Thank you for the question. SAD-Flower augments a flow-matching planner with a virtual control input so that sampled trajectories satisfy safety, admissibility, and dynamic consistency. The generative model—the “drift” term in Eq. (4) of the main paper—is trained purely to imitate expert trajectories, not to optimize a reward. Therefore, sparse or long-horizon rewards do not affect the method; the only requirement is access to demonstration trajectories of a reasonable horizon.
>
> Flow-matching planners are well suited for long-horizon tasks because they generate entire trajectories jointly, as also demonstrated in Diffuser [1]. The model is explicitly trained for the accuracy of its generated full trajectories rather than single-step prediction error, thereby avoiding compounding model errors and scaling more gracefully with respect to long planning horizons. SAD-Flower does not alter the drift dynamics, so it inherits this property: whenever the underlying flow model can produce coherent long-horizon plans, our control augmentation preserves these trajectories while ensuring constraint satisfaction. This is reflected in our Maze2d experiments, where rewards remain high even after enforcing constraints.
>
> Nevertheless, the performance of any generative planner on long‑horizon tasks ultimately depends on the quality of the available training data and the system model. A flow‑matching model trained on short demonstrations may struggle to extrapolate to very long horizons. Recent work on extendable planning via hierarchical multiscale diffusion (HM‑Diffuser)[2] addresses this limitation by progressively extending training trajectories and reasoning across temporal scales; this approach enables diffusion models to generate plans far beyond the lengths seen in data. If such techniques are applied to the base planner, SAD‑Flower can be used on top of it to enforce constraints without retraining.
>
> In summary, flow matching formulation does not use rewards; thus, the criticism regarding “sparse reward” does not apply. Our objective is to match expert trajectories and satisfy constraints, so the presence or absence of reward signals does not change the flow‑matching mechanism. Long-horizon capability is determined primarily by the underlying flow-matching planner and the richness of its demonstrations, while SAD-Flower enforces safety, admissibility, and dynamic consistency on top of whatever trajectories the base model can generate.
>
> *[1] M. Janner, Y. Du, J. B. Tenenbaum & S. Levine, "Planning with Diffusion for Flexible Behavior Synthesis," ICML, 2022.*
> *[2] C. Chen, H. Hany, B. Doojin, K. Taegu, B. Yoshua, A. Sungjin, "Extendable long-horizon planning via hierarchical multiscale diffusion," arxiv, 2025.*
>
>
>
> > ***Q5***: *The conceptual and practical comparison to other hybrid control architectures (e.g., learning-based MPC) is not fully developed, and the distinct pros/cons of SAD-Flower remain unclear.*
>
> **Response:**
>
> Thank you for the question. Hybrid control architectures such as MPC combined with safety filters or formal verification tools share the high-level goal of enforcing constraints, but they differ fundamentally from SAD-Flower in both objective and methodology.
>
> Conceptually, approaches like MPC with learned or analytical models aim to optimize a task-specific cost (e.g., tracking, energy minimization). **They do not attempt to model or reproduce the distribution of expert trajectories.** In contrast, SAD-Flower is a generative planning method: its primary objective is to approximate a trajectory distribution learned from data, while simultaneously enforcing safety, admissibility, and dynamic consistency through a control-augmented flow. Because generative modeling and control optimization solve different problems, MPC-style controllers cannot be expected to replicate the behavior or multimodality of flow matching or diffusion models.
>
> Practically, safety-guaranteed MPC requires repeatedly solving large nonlinear programs, which quickly becomes computationally prohibitive in high-dimensional systems. **SAD-Flower instead solves a lightweight QP, as discussed in the “QP vs. Non-convex Optimization” paragraph of `Section 6.3`**. This results in substantially lower computational cost while still providing constraint satisfaction.
>
> In summary, the strength of SAD-Flower lies in unifying generative modeling with principled control-theoretic guarantees. MPC-based architectures are powerful tools for optimization-based control, but they are not designed to approximate complex trajectory distributions, nor to integrate seamlessly with flow-matching–based sampling, and they typically require repeatedly solving nonlinear optimization problems.

---

> ### Author Response · Authors · 2025-11-24
> **Response to review comments (5/5)**
>
> > ***Q6***: *The potential extension of the framework to multi-agent settings with cooperative or adversarial interactions, and how constraint satisfaction and dynamic consistency guarantees would carry over, is not addressed.*
>
> **Response:**
>
> The current framework is designed for single‑agent trajectories. Extending it to cooperative or adversarial multi‑agent scenarios would require substantial changes to the state representation, constraint formulation and control design. In single‑agent planning, safety and admissibility are expressed as inclusion in sets S and A, and dynamic consistency enforces the dynamics of one system. In a multi‑agent setting, one must define a joint state space that captures the configuration of all agents and encode collision‑avoidance and joint action limits as coupled constraints. The dynamic consistency requirement would then involve the simultaneous evolution of all agents’ states under possibly coupled dynamics, which significantly complicates the Lyapunov‑based design used in our work.
>
> Recent diffusion‑based planners demonstrate that multi‑agent planning is possible but non‑trivial. Multi‑robot Motion Planning with Diffusion Models (MMD) [1] combines single‑robot diffusion models with search‑based techniques to generate collision‑free trajectories for multiple robots; the authors note that multi‑robot diffusion models are difficult due to the high sample complexity and propose using single‑robot models plus classical search for collision constraints. This highlights that extending generative planners to multi‑agent settings usually involves decomposing the problem or adding search layers. Our constraint augmentation could, in principle, be applied to each agent with agent‑specific CBFs and a joint CLF, but the quadratic program would have to enforce inter‑agent collision avoidance and synchronize the dynamics of all agents, which may be computationally heavy. Exploring these extensions is beyond the scope of the current work, and we consider it an interesting direction for future research.
>
> *[1] Y. Shaoul et al., Multi‑Robot Motion Planning with Diffusion Models, International Conference on Learning Representations, 2025.*
>
>
> > ***Q7***: *Have there been any preliminary tests or considerations for deploying SAD-Flower on real-world robotic platforms, and what challenges might arise regarding sensor noise, actuation limits, and real-time execution?*
>
> **Response:**
>
> We thank the reviewer for raising this question about real-robot deployment. While our evaluations focus on simulated navigation, locomotion, and manipulation benchmarks, we note that flow- and diffusion-based models have been demonstrated on physical robotic systems in prior work [3,4]. These results indicate that generative trajectory models can be integrated into standard robotic pipelines involving perception, state estimation, and trajectory execution. We have added a discussion of these considerations in `Appendix K` of the revised manuscript.
>
> SAD-Flower operates purely at the planning level: it generates a full trajectory before execution, rather than running as a real-time feedback controller. As a result, strict real-time control rates are not required, and the computational overhead of solving the QP during sampling does not pose a fundamental obstacle. Moreover, recent advances in efficient diffusion planning—including DiffuserLite [1] and Habi [2]—can be combined with our framework to significantly accelerate trajectory generation. These developments suggest that even tighter planning budgets can be achieved with minimal modification to our formulation.
>
> Overall, given existing real-robot deployments of diffusion and flow-matching planners and the moderate computational cost of our QP-augmented sampling, we believe SAD-Flower can be incorporated into practical generative-planning pipelines with limited adaptation, while providing additional safety, admissibility, and dynamic-consistency guarantees.
>
>
> *[1] Z. Dong, J. Hao, Y. Yuan, F. Ni, Y. Wang, P. Li, and Y. Zheng, “Diffuserlite: Towards real-time diffusion planning,” Advances in Neural Information Processing Systems, vol. 37, pp. 122 556–122 583, 2024.*
>
> *[2] Haofei Lu, Yifei Shen, Dongsheng Li, Junliang Xing, and Dongqi Han. “Habitizing diffusion planning for efficient and effective decision making,” ICML, 2025*
>
> *[3] C. Chi, S. Feng, Y. Du, Z. Xu, E. Cousineau, B. Burchfiel, and S. Song,
> “Diffusion policy: Visuomotor policy learning via action diffusion,” in RSS, 2023.*
>
> *[4] A. Sochopoulos, N. Malkin, N. Tsagkas, J. Moura, M. Gienger, and S. Vijayakumar, “Fast flow-based visuomotor policies via conditional optimal transport couplings,” arXiv:2505.01179, 2025.*

---

### Official Review · Reviewer_MwWH · 2025-10-31

**Soundness:** 2
**Presentation:** 3
**Contribution:** 2
**Rating:** 4
**Confidence:** 3

**Summary:**

This paper proposes SAD-Flower, a method to enforce constraints on trajectories generated by flow matching (FM) models. The authors aim to guarantee safety (state constraints), admissibility (action constraints), and dynamic consistency. The core idea is to augment the standard FM ODE with a virtual control input $u_t$, which is solved for at test time using a Quadratic Program (QP). This QP is formulated to satisfy conditions derived from Control Barrier Functions (CBFs) for safety and admissibility, and a Control Lyapunov Function (CLF) for dynamic consistency. The method is shown to enforce constraints more effectively than several baselines on a few benchmark tasks.

**Strengths:**

1. The paper addresses the important and practical problem of constraint enforcement in generative planning.

2. The proposed method outlines three key aspects in trajectory planning, namely safety, admissibility, and dynamic consistency, and combines Control Barrier Constraints, Control Lyapunov Constraints, and Constrained Minimum-Norm Optimal Control to achieve constraint-aware generative planning.


3. The empirical results show that the proposed method does achieve better constraint satisfaction than the chosen baselines, especially under stricter test-time constraints.

**Weaknesses:**

1. The novelty of the paper seems marginal. The idea of using control-theoretic guidance during the sampling process of a generative model is already established. Specifically, SafeDiffuser [1], cited by the authors, already introduced the core concept of using CBFs to project and guide the sampling steps of a diffusion model to enforce state constraints, also CoBL-Diffusion[2] have introduced control lyapunov functions to diffusion planning. Therefore, the paper appears to be a straightforward combination of Flow Matching + CBF/CLF control.

2. The authors assumes the feasibility of QP. Theorem 5.1 is entirely conditional on the QP (Eq. 5) being feasible at every integration step. In robotics and control, QP infeasibility (e.g., due to conflicting constraints or a poor reference $v_t^\theta$) is a well-known and common failure mode. The paper offers no robust fallback strategy (e.g., constraint relaxation, slack variables) and fails to analyze what happens when this core assumption is violated.

3. The proposed method is heavily dependent on an accurate dynamics model, The entire guarantee for dynamic consistency relies on the CLF $V(\tau_t)$, which requires explicit knowledge of the system dynamics $f(s,a)$ and its derivatives. The authors state this model "can be learned", but this undermines the claim of a "guarantee." The method hence might not guarantee true dynamic consistency; it only guarantees consistency with respect to a learned, approximate model $\hat{f}$. If the learned model is suboptimal, the additional guidance might even decrease the quality of the generated trajectory.

4. Given the method's critical dependence on a learned dynamics model $\hat{f}$, the paper should provide some form of sensitivity analysis or ablation on the model accuracy and how it affects the planning performance.

**Questions:**

1. Could the authors please clearly differentiate their contribution from a straightforward application of standard CLF-CBF-QP control to a flow-matching vector field, especially in light of prior work like SafeDiffuser that already established CBF-guided generative planning?


2. CoBL-Diffusion[2] seems like a closely related work to the proposed method, which also leverage control barrier and lyapunov functions. Authors do mention this work in the related work and state that their "formal guarantees are missing, and action constraints are not addressed". However, while with formal guarantees, from the experiment results, SAD-Flower will still violate the constraints, especially in higher dimension agent state space like the Walker2D tasks. In addition, this method seems like an important baseline and the authors do not compare with it.


3. For the Dynamics Model, since the dynamic consistency "guarantee" is only with respect to a learned model $\hat{f}$, how can this be considered a formal guarantee of true physical consistency? What is the empirical performance (consistency violation) when evaluated against a ground-true dynamics $f$ (if possible), and how does this degrade with the accuracy of $\hat{f}$? Is it easy to obtain a good $f$ in a more general setting?


4. By definition in the paper (e.g., in line 238), obtaining the left-hand side of Equation (CBF-s), (CBF-a), (CLF) requires differentiability. Can the proposed method work when these equations are not differentiable?


5. What were the exact planning horizons ($H$) used for all experiments in Table 1? How does the QP solve time actually scale with $H$ and the dimension of the agent's state space?




[1] SafeDiffuser: Safe Planning with Diffusion Probabilistic Models
[2] CoBL-Diffusion: Diffusion-Based Conditional Robot Planning in Dynamic Environments Using Control Barrier and Lyapunov Functions

---

> ### Author Response · Authors · 2025-11-24
> **Response to review comments (1/5)**
>
> We sincerely thank Reviewer MwWH for the thoughtful and analytically grounded evaluation. We are encouraged by your acknowledgement that the paper addresses a practical and important challenge in constraint-aware generative planning, and by your recognition of the conceptual clarity in unifying safety, admissibility, and dynamic consistency using CBFs, CLFs, and optimal control. We also appreciate your positive remarks on our empirical findings, particularly under tightened constraints. Your feedback brings valuable attention to issues of novelty, feasibility, and reliance on model accuracy, and we provide clarifications on these points in the response below.
>
> > ***W1***: *The novelty appears limited because prior work such as SafeDiffuser and CoBL-Diffusion has already applied CBFs and CLFs to guide generative models, making the method seem like a straightforward combination of Flow Matching with standard CBF/CLF control.*
>
> > ***Q1***: *The paper should more clearly differentiate its contributions from simply applying a standard CLF-CBF-QP controller to a flow-matching vector field, especially given prior CBF-guided generative planning like SafeDiffuser.*
>
> > ***Q2***: *CoBL-Diffusion is a closely related CBF+CLF-based method that is not included as a baseline, even though SAD-Flower still exhibits constraint violations in higher-dimensional settings (e.g., Walker2D), raising questions about comparative performance and the strength of the claimed guarantees.*
>
> **Response:**
>
> We thank the reviewers for raising these thoughtful questions regarding the novelty of our contribution and its relationship to prior work such as SafeDiffuser and CoBL-Diffusion. We appreciate the opportunity to clarify the conceptual and technical distinctions. While these earlier works explore the use of control-theoretic ideas in generative planning, SAD-Flower introduces a set of contributions that go substantially beyond a straightforward combination of flow matching with CBF/CLF constraints. Below, we summarize these differences both at a high level and for each method individually.
>
> ## 1. Core Contribution of SAD-Flower
> At a conceptual level, our work is the first to formulate generative trajectory sampling as a controlled flow-matching system that jointly enforces:
> * state safety via CBFs,
> * action admissibility via CBFs, and
> * dynamic consistency via a dedicated CLF,
>
>
> all enforced at the final time $t=1$ using a prescribed-time schedule.
> This yields a generative planner capable of enforcing entirely new, out-of-distribution constraints during sampling without the need for retraining.
>
> ## 2. High-Level Distinctions From Prior Control-Guided Diffusion Methods
> While prior work applies control ideas to diffusion, existing methods differ fundamentally from SAD-Flower in these fundamental aspects:
> * They do not jointly handle state + action constraints.
> * They do not enforce dynamic consistency.
>
> Furthermore, SafeDiffuser and CoBL-Diffusion rely on very different mechanisms—projection and reward shaping at every step, respectively—that do not guarantee constraint satisfaction, especially under unseen constraints.
>
> ## 3. Differences from SafeDiffuser
> SafeDiffuser also incorporates CBFs, but the two approaches diverge in key aspects:
> ### (a) State–action constraints
> * SAD-Flower provides a unified CBF construction for both state and action constraints.
> * SafeDiffuser considers only state constraints and provides no procedure for action bounds.
> ### (b) Dynamic consistency via a dedicated CLF
> * SAD-Flower is the first to formulate dynamic consistency as a CLF constraint (`Section 5.2`) during generative sampling.
> * SafeDiffuser does not address this requirement and exhibits significant inconsistency in our experiments (Table 1).
>
>
> This distinction is crucial for producing trajectories that are actually executable.
> ### \(c) Prescribed-time enforcement at $t=1$
> * SAD-Flower uses a prescribed-time schedule to enforce all constraints at the final time $t = 1$.
> * SafeDiffuser enforces safety only asymptotically or finite time, which often results in relaxed constraints during sampling.
>
>
> Our experiments in `Section 6.2` show that this prescribed-time schedule yields consistently improved constraint satisfaction.
> (Continue to next response)

---

> ### Author Response · Authors · 2025-11-24
> **Response to review comments (2/5)**
>
> ## 4. Differences from CoBL-Diffusion
> Despite sharing terminology, the two approaches operate in fundamentally different regimes.
> ### (a) CLF/CBF defined on different dynamics
> * CoBL-Diffusion defines CLF and CBF on the robot’s physical dynamics and predicts only actions. Its CLF is used for stability toward the goal, not dynamic consistency. Designing CBF/CLF functions for nonlinear, high-dimensional robot dynamics is challenging and lacks general guidelines.
> * SAD-Flower instead defines both CLF and CBF on the flow-matching dynamics augmented with a virtual control input, which reduces to a simple integrator, making the design structured, scalable, and easier to verify.
> ### (b) Reward shaping vs. hard constraints
> * CoBL-Diffusion uses reward shaping to encourage CLF/CBF behavior, without any guarantee that the constraints will actually be satisfied. Performance depends heavily on manually tuned weights, and CBF/CLF violations can easily persist.
> * SAD-Flower instead enforces CBF and CLF conditions directly through a QP, making constraint handling reliable and predictable.
> ### \(c) Empirical comparison and reproducibility
> CoBL-Diffusion has no public implementation and was evaluated only on simple integrator systems. Nevertheless, **we implemented a CoBL-style diffusion baseline on the LargeMaze task and included it in `Appendix L` in revised paper** with:
> * the same horizon as other baselines,
> * CBF rewards derived from obstacle distances,
> * a CLF based on distance to the goal.
>
>
> **CoBL-Diffusion failed to satisfy both safety and admissibility** (Table A), illustrating the fundamental limitations of reward shaping:
> Table A: Performance of CoBL-Diffusion on large maze task
> | Method | Safety($\downarrow$) | Admissibility($\downarrow$) | Dynamic consistency($\downarrow$) | score($\uparrow$) |
> | -------- | -------- | -------- | -------- | -------- |
> | CoBL-Diffusion     | $0.01\pm0.04$     | $0.23\pm0.33$     | $0.00\pm0.00$ | $0.15\pm0.18$     |
>
> Because CoBL predicts only actions and uses gradients of CBF/CLF rather than solving a constrained problem, it does not provide formal guarantees and cannot enforce action constraints. In addition, the dynamic consistency can be achieved only if a perfect dynamic model is obtained.
> ## 5. Summary
> We appreciate the reviewer's questions regarding novelty and related work. Although SafeDiffuser and CoBL-Diffusion incorporate control-theoretic ideas, SAD-Flower introduces several contributions that are not present in prior work, including:
> * structured CBFs for both state and action constraints,
> * a CLF specifically designed to ensure dynamic consistency,
> * prescribed-time constraint enforcement at $t=1$,
> * a unified QP framework for reliable constraint satisfaction, and
> * the ability to enforce entirely new constraints not present in the dataset.
>
> These elements make SAD-Flower substantially different from and more principled than simply applying existing control filters to a generative model. We hope this clarification helps articulate the novelty and significance of our approach.

---

> ### Author Response · Authors · 2025-11-24
> **Response to review comments (3/5)**
>
> > ***W2***: *The theoretical results rely on the QP being feasible at every step, yet the paper does not address how to handle infeasibility, a common issue when constraints conflict or the reference direction is poor.*
>
> **Response:**
>
> Thank you for raising this concern. While it is correct that the feasibility of the QP is a problem in general when using CBFs and CLFs, this issue is significantly mitigated in our approach.
>
> First, the dynamics for which we design the CBFs and CLF **are not the robot’s physical dynamics, but the learned flow augmented with a virtual control input**. From a control perspective, this corresponds to a simple integrator, which makes the control design considerably more benign than in classical robotic systems.
>
> Second, we provide theoretical conditions under which the individual constraints are guaranteed to be feasible.
> * Theorem B.1 shows that using a signed-distance CBF yields feasible constraints under weak and verifiable assumptions.
> * Theorem B.2 shows that a quadratic CLF is also feasible under similarly mild conditions.
>
> Thus, the individual CBF and CLF constraints are each feasible by construction. QP infeasibility could only arise if the CBF and CLF constraints simultaneously demand control inputs that point in opposite directions. Given the high-dimensional trajectory space and the simplicity of the underlying dynamics, such perfect conflict is highly unlikely in practice. This intuition is supported by our empirical observations: across all experiments, we did not encounter a single infeasible QP, and therefore no fallback strategy was needed. We note that the use of slack variables for softening constraints is already discussed after Theorem 5.1 and can be incorporated when necessary, but it was never required in any of our benchmarks.
>
> Overall, while QP infeasibility is an important issue in classical control settings, our theoretical guarantees and empirical results indicate that it is not a practical concern for SAD-Flower.
>
> > ***Q4***: *By definition in the paper (e.g., in line 238), obtaining the left-hand side of Equation (CBF-s), (CBF-a), (CLF) requires differentiability. Can the proposed method work when these equations are not differentiable?*
>
> **Response:**
>
> Thanks for the question. In control theory, differentiability of the Control Barrier Functions (CBFs) and Control Lyapunov Functions (CLFs) is indeed a standard condition for establishing rigorous guarantees. From a practical perspective, differentiability almost everywhere suffices as illustrated in existing work [1]. Note that the signed distance function that we propose as CBF satisfies this property. We can sometimes even decompose it into multiple CBFs to recover full differentiability as discussed in `Appendix B.1`. Our proposed CLF inherits smoothness from the dynamics of the robot. Thus, it is typically also differentiable.
>
> *[1] Kehan Long, Cheng Qian, Jorge Cort´es, and Nikolay Atanasov. "Learning barrier functions with memory for robust safe navigation." IEEE Robotics and Automation Letters, 6(3):4931–4938, 2021.*
>
>
> > ***Q5***: *The exact planning horizons used in Table 1 are not specified, and there is no analysis of how the QP solve time scales with horizon length or the dimensionality of the agent’s state space.*
>
> **Response:**
>
> Thank you for the question. The exact planning horizons $H$ and the average QP solve time per integration step for SAD-Flower are provided below. **We have included this information in `Appendix I` of the revised paper.**
>
> Table B: Computational cost and settings for SAD-Flower
>
> | Metric | UMaze | LargeMaze | Hopper(Med-Expert) | Hopper(Medium) | Walker2d(Med-Expert) | Walker2d(Medium) | KUKA Block Stacking |
> |:------:|:-----:|:----------:|:-------------------:|:----------------:|:----------------------:|:------------------:|:---------------------:|
> | Horizon | $128$ | $384$ | $50$ | $50$ | $50$ | $50$ | $128$ |
> | State Dim | $4$ | $4$ | $11$ | $11$ | $17$ | $17$ | $39$ |
> | Act Dim | $2$ | $2$ | $3$ | $3$ | $6$ | $6$ | $7$ |
> | Time(s) | $0.09 \pm 0.02$ | $0.13 \pm 0.01$ | $0.06 \pm 0.01$ | $0.12 \pm 0.01$ | $0.09 \pm 0.01$ | $0.09 \pm 0.01$ | $0.74 \pm 0.08$ |
>
>
> The numbers reveal only a mild growth in computation time as the problem complexity increases. For example, comparing UMaze (H = 128) and LargeMaze (H = 384) shows that a 3× increase in horizon leads to only a ~40 ms increase in QP solve time. Likewise, moving from low-dimensional navigation (4D state, 2D action) to high-dimensional manipulation (39D state, 7D action) increases the cost to ~0.74 s, which remains manageable given the significantly higher dimensionality.

---

> ### Author Response · Authors · 2025-11-24
> **Response to review comments (4/5)**
>
> > ***W3, W4***: *The method’s guarantees of dynamic consistency rely on a learned dynamics model and its derivatives, so in practice it only guarantees consistency with respect to an approximate model rather than the true system; without sensitivity analysis or ablations, the impact of model error on planning quality remains unclear.*
>
> > ***Q3***: *Since the dynamics model is learned, it is not obvious how the claimed guarantees translate to true physical consistency, how consistency violations behave under the real dynamics as model accuracy degrades, or how feasible it is to obtain a sufficiently accurate model in more general settings.*
>
> **Response:**
>
> We thank the reviewer for these insightful questions regarding the dependence of SAD-Flower on model accuracy and the interpretation of our dynamic-consistency guarantees. We appreciate the opportunity to clarify this point.
>
> We would like to clarify that our formal guarantee does not rely on a learned dynamics model. All theoretical results in `Section 4` and `Appendix B` assume access to an accurate dynamics function $\boldsymbol{f}(\boldsymbol{s}(k), \boldsymbol{a}(k))$, and under this assumption, our CLF construction provides a rigorous guarantee of dynamic consistency. This requirement is realistic for some robotic systems, where high-fidelity analytical simulators and physics engines provide reliable models [1–3].
>
> When no analytical model is available, the dynamics can instead be learned from data. In this case, the guarantee naturally becomes approximate: the CLF enforces consistency with respect to the learned model $\hat{\boldsymbol{f}}$. However, this does not invalidate the guarantee—it simply ties its accuracy to that of the learned model. The relationship between model error and consistency error is well understood in model-based reinforcement learning (e.g., [4]), where it is established that small model error leads to proportionally small deviations in rollouts. In other words, practical dynamic consistency is still achieved as long as the learned model is reasonably accurate.
>
> To empirically verify this intuition, **we include in `Section 6.4` (revised paper) an ablation study** in which we purposely **degrade model accuracy** by training the forward dynamic model on progressively smaller subsets of the Hopper-Medium dataset. As shown in Table B below, SAD-Flower maintains perfect safety and admissibility across all reduced-data settings down to $10\%$, and dynamic consistency remains stable. Degradation becomes apparent only when the dataset is reduced to 0.01%, at which point safety violations emerge. This confirms that our method is robust to realistic levels of model approximation and that obtaining an accurate enough model is feasible in practice.
>
> Finally, we emphasize that **only the dynamic-consistency CLF depends on the dynamics model**. The safety and admissibility guarantees—enforced via CBFs—remain exact and do not rely on model accuracy.
>
> Table B: Effect of training dataset size on dynamic consistency and constraint satisfaction (Hopper-Medium).
>
> | Dataset | Safety ($\downarrow$) | Admissibility ($\downarrow$) | Dynamic consistency ($\downarrow$) | score ($\uparrow$) |
> |--------|------------------------|-------------------------------|------------------------------------|---------------------|
> | 100%   | $0.00 \pm 0.00$        | $0.00 \pm 0.00$               | $0.01 \pm 0.01$                    | $0.34 \pm 0.01$     |
> | 90%    | $0.00 \pm 0.00$        | $0.00 \pm 0.00$               | $0.01 \pm 0.01$                    | $0.35 \pm 0.01$     |
> | 80%    | $0.00 \pm 0.00$        | $0.00 \pm 0.00$               | $0.01 \pm 0.01$                    | $0.34 \pm 0.02$     |
> | 70%    | $0.00 \pm 0.00$        | $0.00 \pm 0.00$               | $0.01 \pm 0.01$                    | $0.33 \pm 0.04$     |
> | 60%    | $0.00 \pm 0.00$        | $0.00 \pm 0.00$               | $0.01 \pm 0.01$                    | $0.39 \pm 0.03$     |
> | 50%    | $0.00 \pm 0.00$        | $0.00 \pm 0.00$               | $0.01 \pm 0.01$                    | $0.35 \pm 0.07$     |
> | 40%    | $0.00 \pm 0.00$       | $0.00 \pm 0.00$               | $0.01 \pm 0.01$                    | $0.38 \pm 0.06$     |
> | 30%    | $0.00 \pm 0.00$       | $0.00 \pm 0.00$               | $0.01 \pm 0.01$                    | $0.39 \pm 0.07$     |
> | 20%    | $0.00 \pm 0.00$       | $0.00 \pm 0.00$               | $0.01 \pm 0.01$                    | $0.38 \pm 0.02$     |
> | 10%    | $0.00 \pm 0.00$       | $0.00 \pm 0.00$               | $0.01 \pm 0.01$                    | $0.38 \pm 0.03$     |
> | 0.01%    | $0.02 \pm 0.09$       | $0.00 \pm 0.00$               | $0.04 \pm 0.02$                    | $0.36 \pm 0.03$     |
>
> (Reference is attached in the next response due to word limit)

---

> ### Author Response · Authors · 2025-11-24
> **Response to review comments (5/5)**
>
> *[1] C. Gaz, M. Cognetti, A. Oliva, P. R. Giordano, and A. De Luca, “Dynamic identification of the franka emika panda robot with retrieval of feasible parameters using penalty-based optimization,” IEEE Robotics and Automation Letters, 4(4), 4147-4154, 2019.*
>
> *[2] T. A. Howell, S. L. Cleac'h, J. Brüdigam, J. Z. Kolter, M. Schwager, and Z. Manchester, “Dojo: A differentiable physics engine for robotics,” arXiv preprint arXiv:2203.00806, 2022.*
>
> *[3] B. Acosta, W. Yang, and M. Posa, "Validating robotics simulators on real-world impacts," IEEE Robotics and Automation Letters 7.3: 6471-6478, 2022.*
>
> *[4] C., Sebastian, F. Berkenkamp, and A. Krause. "Efficient model-based reinforcement learning through optimistic policy search and planning." Advances in Neural Information Processing Systems, 14156-14170, 2020.*

---

### Official Review · Reviewer_DCoi · 2025-11-01

**Soundness:** 3
**Presentation:** 3
**Contribution:** 2
**Rating:** 4
**Confidence:** 2

**Summary:**

The paper uses Flow Matching to learn a model for planning in maze and robotics domains. They propose to modify regular flow matching to include a control term. This control term is designed so that once the FM process is run for sufficient iterations, the resultant should not violate certain safety, feasibility and dynamics constraints. The algorithm is evaluated on maze and robotics domains, where a model is trained from expert data as well as to stay within constraints. Comparisons with other generative models shows almost perfect adherence to constraints while maintaining high performance.

**Strengths:**

1. The results demonstrate that including the control terms proposed in the paper leads to solutions that better adhere to constraints across three different tasks.

2. The paper combines ideas from generative modeling and control theory in an interesting way to demonstrate practical results.

Overall the paper demonstrates an interesting way to incorporate reasoning about hard constraints into generative modeling for planning.

**Weaknesses:**

The paper frames the modeling of trajectories together with various contraints as planning. One of the advantages of classical planning approaches to the sorts of tasks presented in the paper (RRT style algorithms for low level problems, PDDL for bi-level planning) is their generalisability to novel settings without the need for re-training. Will using Flow Matching based methods as "planners" provide this flexibility?

Overall, it is not clear to me whether such an approach benefits downstream planning tasks. Since experts (such as navigation algorithms) are being used to generate training data for the model, how does the final performance of FM compare to running the expert directly on the task?

The evaluation of this approach is also limited to mazes and a single robotics application. The approach may have potential towards being applied to complex grasping, where the additional constraint control term helps generate valid grasps in complex environments. Such additional settings are not evaluated.

**Questions:**

1. Is the transition function known for the domains being evaluated? Specifically for the Kuka robotics task? Is this method applicable to domains where such a function would be computationally difficult to evaluate (e.g. stochastic or complex environments with passive dynamics)?

2. Does the block stacking task consider cluttered environments where the system has to reason about difficult grasps / avoiding collision with blocks?

3. How much data is used to learn the model?

4. Does the training data include constraints violations? If not, then why does the learned model exhibit constraint violations?

5. How does this compare to a baseline where you sample from a model and reject violating samples?

---

> ### Author Response · Authors · 2025-11-24
> **Response to review comments (1/4)**
>
> We appreciate Reviewer DCoi’s constructive and supportive assessment. Your review highlights the practical impact of coupling flow matching with control-theoretic guidance, and we are glad that the improved constraint adherence across maze navigation, locomotion, and robotic manipulation resonated with you. We also value your recognition that our formulation offers a concrete way to bring hard-constraint reasoning into generative modeling while still preserving strong task performance. The questions you raise regarding generalisation, comparison to classical planners, and applicability to richer robotic settings are especially helpful, and we address them in detail below.
>
>
> > ***Weakness***: *The evaluation covers only maze navigation and a single robotics task, without testing more complex domains such as grasping or cluttered manipulation.*
> >
> > ***Q2***: *Does the block-stacking setup include cluttered scenes that require reasoning about grasp difficulty and collision avoidance?*
>
> **Response:**
>
> We appreciate the reviewer’s insightful comment regarding the scope of our evaluation. We fully agree that demonstrating the applicability of our method in more complex manipulation tasks is valuable. In our experiments, the block stacking environment strictly follows the configuration used in Diffuser [1]. **This setting does not involve additional clutter beyond the blocks being stacked, and we adopt it to ensure comparability with prior work on generative planners**.
>
> To demonstrate the applicability of our method to more complex manipulation scenarios that involve challenging grasps and contact-rich dynamics, we **additionally evaluate on the D4RL Adroit Relocate dexterous grasping task (39-dimensional state, 30-dimensional action).**  We define the state constraint by the arm's position for obstacle avoidance, and the action constraint by the actuator limits. This high-dimensional, contact-rich manipulation scenario represents a substantially more challenging benchmark than the tasks previously considered. The full experimental setup and results have been **added to `Section 6.4` in the revised paper.**
>
> The results (see table below) highlight the core motivation of SAD-Flower: while standard flow matching (FM) achieves reasonable task performance, it exhibits notable violations of safety and admissibility constraints due to its unconstrained sampling. In contrast, **SAD-Flower reliably enforces both state and action constraints**, even in this challenging setting. Although dynamical consistency is not perfectly satisfied, the residual violations are small and primarily attributable to discretization and numerical integration.
>
> This experiment demonstrates that our method scales effectively to complex grasping domains and provides evidence that the proposed control-augmented mechanism meaningfully improves the reliability and safety of generative planners in high-dimensional robotic applications.
>
> Table A:  Performance of SAD-Flower and FM in the dexterous grasping scenario
>
> | Method | Safety($\downarrow$) | Admissibility($\downarrow$) | Dynamic consistency($\downarrow$) | score($\uparrow$) |
> | -------- | -------- | -------- | -------- | -------- |
> | FM     | $0.15 \pm 0.21$     | $0.62 \pm 0.19$     | $0.07 \pm 0.04$ | $1.07 \pm 0.08$     |
> | SAD-Flower     | $\boldsymbol{0.00 \pm 0.00}$     | $\boldsymbol{0.00 \pm 0.00}$     | $\boldsymbol{0.06 \pm 0.09}$ | $1.05 \pm 0.05$     |
>
> > ***Q3***: *How much data is required to train the model?*
>
> **Response:**
>
> The dataset specifications are detailed in `Section 6.1` and `Appendix D` of the paper. A brief summary is as follows:
>
> * **Maze2D**
>     * *Umaze*: **1 million trajectories** (D4RL)
>     * *Large*: **4 million trajectories** (D4RL)
>
> * **Hopper**
>     * *Medium*: **1 million trajectories** (partially trained SAC)
>     * *Medium-Expert*: **2 million trajectories** (mixture of expert and partial SAC)
>
> * **Walker2D**
>     * *Medium*: **1 million trajectories** (partially trained SAC)
>     * *Medium-Expert*: **2 million trajectories** (mixture of expert and partial SAC)
>
> * **Kuka Block-Stacking**
>     * **10,000 trajectories** (generated by PDDLStream)
>
> All datasets used in our experiments are **public, open-sourced**, and commonly used in the offline RL/diffusion-planning literature.

---

> ### Author Response · Authors · 2025-11-24
> **Response to review comments  (2/4)**
>
> > ***Q1***: *Is the transition function known in each domain, especially the Kuka task, and can the approach handle environments where evaluating the transition dynamics is difficult or stochastic?*
>
> **Response:**
>
> Our problem formulation in Section 3 models the system as a general discrete-time nonlinear dynamics $\boldsymbol{s}(k+1)=\boldsymbol{f}(\boldsymbol{s}(k),\boldsymbol{a}(k))$. SAD-Flower does not require analytic access to $\boldsymbol{f}$. Instead, it only needs an estimate of $\boldsymbol{f}$ to evaluate the CLF condition during the controlled sampling phase in Section 5.2. When the true transition function is unknown, we learn a forward dynamics model from the same offline trajectory dataset used to train the FM model. This design is already described in the CLF paragraph in `Section 5.2`, where we note that $\nabla V$ is computed using a learned model when $\boldsymbol{f}$ is unknown. Across the evaluated domains, this works as follows:
>
> **Maze2D.** Maze2D uses a simple double integrator for a point mass. The environment dynamics are described analytically in Appendix F of the paper, and we also train a forward model for use inside the constraint-enforcement layer. Because the system is low-dimensional, having the exact model is not restrictive.
>
> **Hopper and Walker2D.** In these tasks, we have access to a MuJoCo simulator. Thus, a transition model exists and can be queried. Nevertheless, as stated in Section 5.2 and evaluated in `Section 6.2`, we learn a neural forward dynamics model from data and use this learned model inside the CLF constraint. This choice demonstrates that SAD-Flower does not rely on knowing the exact simulator dynamics and remains applicable whenever only offline data are available.
>
> **Kuka block stacking.** This task is simulated in PyBullet. The planner generates joint-space trajectories conditioned on block positions. Only state constraints related to self-collision avoidance are enforced, and no action or dynamic consistency constraints are activated for this task, as specified in `Section 6.1`.
>
> **General applicability:** The control-theoretic layer requires an estimate of $\boldsymbol{f}$ to define the CLF. When the environment has unknown or moderately stochastic dynamics, the learned forward model can still be used, as done in Maze, Hopper, and Walker2D.
>
> To directly address the reviewer’s concern about the evaluation on difficult dynamics, we additionally evaluate on the D4RL Adroit Relocate dexterous grasping task (39D dimensional state, 30D dimensional action) and have included it in `Section 6.4` in revised paper. We define the state constraint by the arm's position for obstacle avoidance, and the action constraint by the actuator limits. The result (Table A in part I response) is mentioned in the previous question that SAD-Flower robustly enforces safety and admissibility constraints, whereas standard flow matching exhibits frequent violations. This highlights the **effectiveness of our control-augmented formulation in high-dimensional grasping domains**.

---

> ### Author Response · Authors · 2025-11-24
> **Response to review comments  (3/4)**
>
> > ***Weakness***: *Will Flow Matching used as a planner retain the generalisability of classical methods such as RRT or PDDL, which do not require retraining for new settings?*
> >
> > *How does the performance of the Flow Matching planner compare to running the expert policy directly, given that expert rollouts are used for training?*
>
> **Response:**
>
> Thanks for the question. Classical planners such as RRT or PDDL are valued for generalizing to new settings without retraining. We agree that the ability to reuse a planner without retraining is essential, and the prior work already demonstrates that generative trajectory models can retain this flexibility. In particular, Diffuser [1] provides a clear empirical precedent: although the Maze2D training data “is undirected – consisting of a controller navigating to and from randomly selected locations”, the authors show that a single pretrained trajectory model can be reused across different goals without finetuning, stating that “we do not need to retrain the model from the single-task experiments and simply change the conditioning goal” and that “Diffuser performs as well in the multi-task setting as in the single-task setting.” This demonstrates that a generative trajectory model can generalize to novel task specifications through test-time conditioning alone, without modifying model parameters. Our method follows the same principle. **Flow matching learns a trajectory prior, and SAD-Flower’s test-time controller enforces safety, admissibility, and dynamic consistency constraints without retraining**.
>
>
> Regarding the reviewer’s concern about the benefit of running the expert directly: the generative-model-based method, e.g., Diffuser [1], explicitly shows that generative planners can outperform the data-collection expert, reporting that “Diffuser achieves scores over 100 in all maze sizes, indicating that it **outperforms a reference expert policy**”. This is because the “expert” is not a task-optimal planner but the simple waypoint-following controller used only to populate the dataset. Similar to Diffuser, FM learns long-horizon structure beyond what such controllers optimize for, and SAD-Flower further improves on this by ensuring reliable behavior under constraints unseen during training. Moreover, unlike the SAD-Flower we proposed, **the expert policy itself offers no safety, admissibility, or dynamic consistency guarantees**, **whereas our test-time planner ensures these properties by construction**.  As shown across different tasks in Table 1, SAD Flower achieves competitive reward performance relative to FM and diffusion-based baselines while simultaneously ensuring perfect safety and admissibility, which the original experts do not provide.
>
> Overall, FM-based planners with SAD-Flower do not simply imitate expert demonstrations. They recover a flexible, reusable trajectory prior that can outperform the underlying data-collection experts and remain robust under novel task and constraint settings, achieving a level of adaptability that classical planners provide and extending it with principled safety and dynamic consistency guarantees. In addition, prior work such as Diffuser has already demonstrated the benefit of generative-model-based planners over the expert, and **our contribution in SAD Flower is to show that such planners can also satisfy safety, admissibility, and dynamic consistency**.
>
>
> [1] M. Janner, Y. Du, J. B Tenenbaum, and S. Levine. "Planning with diffusion for flexible behavior synthesis." International Conference on Machine Learning, 2022.

---

> ### Author Response · Authors · 2025-11-24
> **Response to review comments (4/4)**
>
> > ***Q4***: *Does the training set include constraint violations, and if not, why does the learned model still produce violating trajectories?*
>
> **Response:**
>
> We thank the reviewer for raising this helpful question. To clarify: **yes, the training datasets contain trajectories that violate the constraints used during evaluation**, because the stricter constraints are introduced only at test time. As detailed in `Sections 6.1–6.3`:
> * The Maze2D demonstrations are collected without the two superellipse obstacles and with looser control limits.
> * The locomotion datasets do not include the low-ceiling constraints used at evaluation.
> * The Kuka dataset does not include the tightened joint limits imposed during testing.
>
> Thus, the expert trajectories respect the default simulator constraints but **naturally violate the stricter, unseen constraints** applied during evaluation. Because the flow-matching model is trained purely by imitation, it faithfully reproduces expert-style trajectories—even when these demonstrations violate the evaluation-time constraints. Therefore, standard FM exhibits safety and dynamic-consistency violations in Table 1 **not** because the learned model behaves poorly, but simply because it was never trained on data that satisfied those stricter constraints.
>
>
> > ***Q5***: *How does this compare to a baseline where you sample from a model and reject violating samples?*
>
> **Response:**
>
> We thank the reviewer for raising this thoughtful question regarding the use of a reject-sampling baseline to enforce safety and admissibility. A reject-sampling baseline **cannot reliably enforce safety, admissibility, or dynamic consistency** in our setting because the constraints we impose at test time **are novel and unseen during training, as we mentioned in the response to Q4**. Even though flow matching can represent multimodal behavior patterns, **it cannot generate samples that lie inside constraint sets it has never observed**. As a result, the feasible set has extremely low probability mass under the model, making rejection-based filtering impractical.
>
> We evaluate this idea empirically by implementing a reject-sampling baseline across all our benchmarks, including maze navigation, locomotion (Hopper and Walker2d), and the KUKA block stacking task. We train a flow matching model on the original dataset, which includes trajectories that violate test-time constraints, since those constraints are unseen during training. At inference, it samples a batch of trajectory candidates and discards those violating constraints, selecting from the remaining set the trajectory with the least violation. **This baseline and its experimental details are now included in `Appendix L` of the revised paper.**
>
> The results (Table B) show **consistent constraint violations across all environments**. Safety and admissibility remain non-zero in every case, and dynamic-consistency errors persist. These findings confirm that reject sampling cannot reliably enforce unseen constraints, **even when many samples are drawn**. This highlights a key advantage of SAD-Flower: **constraint satisfaction is built into the sampling process** itself through control-augmented flow matching, enabling robust enforcement of safety and admissibility even under entirely novel constraints.
>
> Table B: Performance of the reject-sampling baseline across all benchmarks.
> **Positive values** in safety and admissibility indicate **constraint violations**.
>
> | Experiment | Safety($\downarrow$) | Admissibility($\downarrow$) | Dynamic consistency($\downarrow$) | score($\uparrow$) |
> | -------- | -------- | -------- | -------- | -------- |
> | UMaze     | $0.04 \pm 0.09$     | $0.91 \pm 0.01$     | $0.01 \pm 0.01$ | $2.91 \pm 0.65$     |
> | LargeMaze     | $0.16 \pm 0.29$      | $0.90 \pm 0.01$     | $0.02 \pm 0.01$| $1.52 \pm 0.28$     |
> | Hopper(Med-Expert)     | $0.14 \pm 0.18$    | $0.05 \pm 0.02$     | $0.02 \pm 0.01$ | $0.49 \pm 0.33$     |
> | Hopper(Medium)     | $0.03 \pm 0.04$      | $0.06 \pm 0.03$     | $0.02 \pm 0.02$| $0.24 \pm 0.14$     |
> | Walker2d(Med-Expert)     | $0.08 \pm 0.10$ | $0.08 \pm 0.08$ | $0.04 \pm 0.06$ | $0.97 \pm 0.22$    |
> | Walker2d(Medium)     | $0.18 \pm 0.15$ | $0.16 \pm 0.10$  | $0.07 \pm 0.17$| $0.68 \pm 0.19$   |
> | KUKA Block Stacking  | $0.01 \pm 0.01$ | $-$  | $-$ | $0.44 \pm 0.67$   |

---

### Author Response · Authors · 2025-11-29
**Summary of Author Response**

Dear AC and reviewers,

We sincerely thank all reviewers for their thoughtful comments. **We have prepared a comprehensive response addressing all reviewer comments.** Unfortunately, due to the OpenReview issue, **we did not have the opportunity to receive further reviewer feedback or engage in follow-up discussion.** We would be grateful if the AC could carefully **consider our detailed responses when evaluating the revision, as we are confident that all major points raised by the reviewers have been fully addressed.** These revisions further clarify the main contribution of our paper: **a unified, theoretically grounded, and practically robust flow-matching planner that ensures safety, admissibility, and dynamic consistency—even under novel constraints at test time**.

Below, we provide a summary of the strengths highlighted by the reviewers, the revisions made to the manuscript, and key clarifications offered in our response.

---

## 1. Strengths Highlighted by Reviewers

Reviewers acknowledged several positive aspects of our work, including:
- A novel generative modeling approach with control theory, supported by a rigorous CBF/CLF-based theoretical foundation.  \
(Reviewers DCoi, ri9Z, KkJf)
- The ability to enforce new or tightened constraints without retraining. \
(Reviewers MwWH, ri9Z)
- Comprehensive experiments show that our method reliably satisfies constraints across diverse tasks. \
(Reviewers DCoi, MwWH, ri9Z)
- Clear motivation and structure around the three core objectives—safety, admissibility, and dynamic consistency. \
(Reviewers MwWH, KkJf)

---

## 2. Revisions in the Manuscript

In response to reviewer feedback, we made the following revisions (all highlighted in blue in the updated manuscript):

1. **Scalability to Higher-Dimensional Systems** (Reviewers DCoi, ri9Z, Kkjf)
`Section 6.4 (Table 3)` adds results on **D4RL Adroit Relocate** (dexterous grasping with 39D state, 30D action), demonstrating scalability to high-dimensional, contact-rich manipulation.

2. **Computation-Time Analysis**  (Reviewers MmWH, ri9Z, Kkjf) \
`Appendix I (Table 13)` reports QP solve times across all benchmarks with a discussion of scaling behavior.

3. **Dynamics-Model Accuracy Study**  (Reviewer MmWH) \
`Section 6.4 (Table 5)` investigates degraded dynamics models, showing robustness down to $10\%$ of training data.

4. **Hyperparameter Sensitivity**  (Reviewer ri9Z) \
   `Section 6.4 (Table 4)` analyzes the prescribed-time constant $c$, showing stable performance over a wide range.

5. **Comparison with Related Work (CoBL-Diffusion)**  (Reviewer MmWH) \
   `Appendix L (Table 15)` includes a CoBL-style baseline on LargeMaze, which fails to satisfy constraints, highlighting the limitations of reward shaping.

6. **Comparison with Reject-Sampling Baseline**  (Reviewer DCoi) \
   `Appendix L (Table 14)` evaluates reject sampling across all benchmarks and shows it cannot satisfy safety/admissibility due to unseen test-time constraints.

7. **Expanded Discussion of Related Work**  (Reviewer Kkjf) \
   `Appendix J` clarifies differences from recent CBF/CLF-guided diffusion planners, especially regarding unseen constraints.

---

## 3. Key Clarifications Provided in Text Responses

Beyond manuscript revisions, we addressed several important conceptual questions:

- **Generalisability vs. classical planners:**  (Reviewer DCoi) \
  FM-based planners can generalize to new goals and settings without retraining, as demonstrated in prior work (e.g., Diffuser), and SAD-Flower inherits this property while adding principled constraint enforcement.

- **Novelty relative to SafeDiffuser and CoBL-Diffusion:**  (Reviewer MmWH) \
  We clarified that prior methods
  (i) do not jointly handle state + action constraints,
  (ii) do not enforce dynamic consistency,
  (iii) rely on projection or reward shaping rather than hard constraints, and
  (iv) do not use prescribed-time enforcement.
  SAD-Flower is the first unified framework ensuring all these properties with theoretical support.

- **QP feasibility:**  (Reviewer MmWH) \
  We explained why QP infeasibility is unlikely due to the simple integrator-like structure of FM dynamics and showed that individual CBF/CLF constraints are provably feasible under mild conditions. Empirically, no infeasible QP was ever observed.


We thank all reviewers for their valuable feedback and for helping us improve SAD-Flower.

---

### Meta-Review · Area_Chair_bF7j · 2026-01-08

**Summary:**

Reviewers praised:
- Focus on important problem of constraint enforcement in generative planning.
- Interesting combination of ideas from generative modeling and control theory.

Reviewers were concerned about:
- Placement of the work within the literature
- Lack of novelty / marginal novelty in comparison to methods such as CoBL-Diffusion and SafeDiffuser
- Scalability to higher dimensions
- Lack of runtime analyses

**Reviewer Concerns:**

Addressed:
- Scalability to higher dimensions
- Lack of runtime analyses

Outstanding:
- While the authors provided comprehensive answers to reviewers' questions about the placement of the work within the broader literature, as well as comparisons to specific baselines brought up by reviewers, the core contribution of the work appears to remain narrow. The resulting framework is no doubt useful to practitioners in the field, but multiple reviewers asked for clarification of the novelty of this approach in their reviews, indicating difficulty placing the work within the literature and identifying the core contributions.

**Reviewer Scores:**

It is difficult to estimate if scores would change because, while the authors provided answers to most (if not all) reviewer questions, it is difficult to estimate which aspects were most important for determining their initial rating.

---

### Decision · Program_Chairs · 2026-01-26

Reject